JCB Journal of Cell Biology

# Single-molecule lipid biosensors mitigate inhibition of endogenous effector proteins

Victoria L. Holmes[1]*, Morgan M.C. Ricci[1]*, Claire C. Weckerly[1], Michael Worcester[1], and Gerald R.V. Hammond[1]

**Genetically encoded lipid biosensors uniquely provide real time, spatially resolved kinetic data for lipid dynamics in living cells. Despite clear strengths, these tools have significant drawbacks; most notably, lipid molecules bound to biosensors cannot engage with effectors, potentially inhibiting signaling. Here, we show that although PI 3-kinase (PI3K)–mediated activation of AKT is not significantly reduced in a cell population transfected with a PH-AKT1 $PIP_3/PI(3,4)P_2$ biosensor, single cells expressing PH-AKT at visible levels have reduced activation. Tagging endogenous AKT1 with neonGreen reveals its EGF-mediated translocation to the plasma membrane. Co-transfection with the PH-AKT1 or other $PIP_3$ biosensors eliminates this translocation, despite robust recruitment of the biosensors. Inhibition is even observed with $PI(3,4)P_2$-selective biosensor. However, expressing lipid biosensors at low levels, comparable with those of endogenous AKT, produced no such inhibition. Helpfully, these single-molecule biosensors revealed improved dynamic range and kinetic fidelity compared with overexpressed biosensor. This approach represents a noninvasive way to probe spatiotemporal dynamics of PI3K signaling in living cells.**

## Introduction

Genetically encoded lipid biosensors are a transformative tool for studying the role of specific lipids in cellular physiology (Yang et al., 2018; Wills et al., 2018). They are uniquely able to report both the organellar location of cytosolic leaflet-localized lipids as well as their dynamic changes in intact, living cells. Although a mainstay of experimental techniques for interrogating lipid function, these biosensors, like all experimental tools, have specific drawbacks and limitations. One of the most prominent is the fact that lipid engagement by a biosensor occludes the lipid's headgroup, blocking its interaction with proteins that mediate biological function. It follows that large fractions of lipid may be effectively outcompeted by the biosensor, inhibiting the associated physiology. We have argued that, in most cases, this is unlikely because the total number of lipid molecules outnumbers expressed biosensors by one to two orders of magnitude (Wills et al., 2018). However, for less abundant lipids, total molecule copy numbers may be in the order of tens to hundreds of thousands, making competition by biosensors a real possibility.

One of the most famous second messenger lipids is phosphatidylinositol 3,4,5-trisphosphate ($PIP_3$). Generated by class I phosphoinositide 3-OH kinases (PI3Ks), $PIP_3$ activates numerous effector proteins involved with stimulating cell metabolism, migration, growth, and survival, along with activation of the immune system (Fruman et al., 2017; Vanhaesebroeck et al., 2021; Madsen and Toker, 2023). Its best-known effector is the serine-threonine kinase, AKT (also known as PKB). This protein is recruited to the membrane and allosterically activated by $PIP_3$ binding to its pleckstrin homology (PH) domain, facilitating phosphorylation of the enzyme at threonine 308 and serine 473 (Truebestein et al., 2021; Bae et al., 2022). The selectivity and affinity of the $PIP_3$ interaction with PH domains of proteins like AKT led to their development as some of the first biosensors for lipid signaling (Venkateswarlu et al., 1998; Watton and Downward, 1999; Várnai et al., 1999; GRAY et al., 1999).

$PIP_3$ is synthesized by 3-OH phosphorylation of the substrate lipid, $PI(4,5)P_2$, of which cells contain ~10 million in their plasma membranes (PMs) (Wills and Hammond, 2022). Yet only 3–5% of $PI(4,5)P_2$ is converted to $PIP_3$ (Stephens et al., 1993), resulting in no more than about 500,000 molecules per cell. PH domain–containing $PIP_3$ effector proteins can be predicted based on sequence comparison to known $PIP_3$ effectors versus non-effectors using a recursive functional classification matrix for each amino acid (Park et al., 2008). Inspecting OpenCell proteome data for the number of these binding proteins predicts ~442,000 $PIP_3$ effector protein copies in a HEK293 cell (Cho et al., 2022). Therefore, $PIP_3$ production is well matched to engage this particular cell type's effector proteins. Translating these numbers into cellular concentrations (assuming 13 pl for a 15-μm spherical HEK293 cell) gives ~500 nM concentration, which is again

[1]Department of Cell Biology, University of Pittsburgh School of Medicine, Pittsburgh, PA, USA.

*V.L. Holmes and M.M.C. Ricci contributed equally to this paper. Correspondence to Gerald R.V. Hammond: ghammond@pitt.edu.

**Rockefeller University Press**
J. Cell Biol. 2025 Vol. 224 No. 3 e202412026



well matched by the 100–500 nM dissociation constant of these effector proteins (Wills et al., 2018). However, now consider that an overexpressed PH domain–based biosensor has been measured at up to 10,000 nM (Xu et al., 2003): these domains could easily dominate binding to $PIP_3$ and thus outcompete endogenous effector proteins, blocking physiological signaling. Indeed, there have been prior reports of overexpressed $PIP_3$ biosensors inhibiting PI3K signaling (Várnai et al., 2005). However, in this case it was unclear how much this was due to sequestration of the lipid versus competition for protein-binding partners of the PH domains in a lipid-dependent manner. Furthermore, it remains possible that cells may respond to biosensor $PIP_3$ competition by increasing $PIP_3$ production by yet-to-be–defined feedback mechanisms. Such feedback has been reported in the case of $PI(4,5)P_2$ biosensors (Traynor-Kaplan et al., 2017).

In this manuscript, we have addressed the question of whether overexpressed biosensors compete for $PIP_3$ binding enough to inhibit effector protein translocation. To this end, we generated gene-edited cell lines incorporating a neonGreen fusion protein driven from a native AKT1 allele in HEK293A cells. We show that a variety of $PIP_3$ effector proteins very efficiently block AKT1 translocation and inhibit signaling, despite robust $PIP_3$ production. This competition can be eliminated by reducing biosensor concentrations (through reduced transfection times and/or using weaker promoters) to reach approximate concentrations of the endogenous effectors, though this requires single-molecule sensitivity. As well as alleviating inhibition of PI3K signaling, biosensors expressed at these low levels show improved dynamic range and report more accurate kinetics than their overexpressed counterparts.

## Results

### Inhibition of AKT activation by PH-AKT1 $PIP_3$ biosensor

$PIP_3$ mediated activation of AKT1 occurs via the initiation of both allosteric and localization-based mechanisms (Truebestein et al., 2021; Bae et al., 2022). Firstly, $PIP_3$ binding to the AKT PH domain disrupts an autoinhibitory interaction with the kinase activation loop. Secondly, the enzyme is localized to the membrane, where it is concentrated with upstream activating kinases PDK1 and TORC2, which phosphorylate and activate the enzyme at residues T308 and S473, respectively (Fig. 1 A). Both mechanisms critically depend on $PIP_3$, so we reasoned that high concentrations of $PIP_3$ biosensor might disrupt AKT1 activation by titrating out the lipid.

To test for such an effect, we overexpressed the most commonly utilized $PIP_3$ biosensor, the isolated PH domain of AKT1. As a control, we used the non-$PIP_3$–binding mutant R25C (Watton and Downward, 1999). The most commonly employed method to assay AKT activation is by immunoblotting for the activated phosphorylated species. Therefore, we transiently transfected HEK293A cells with EGFP control, PH-AKT1-EGFP, or the R25C mutant, then stimulated them for 5 min with a moderate experimental dose of EGF (10 ng/ml). Blotting for pS473 revealed no drastic inhibition of pS473 levels by PH-AKT1 expression (Fig. 1, B and C). However, since transient transfection

produces expression in a subset of cells, with those cells displaying a wide distribution of expression levels, we reasoned that analysis of the whole cell population might be misleading, since this includes untransfected cells or cells with low expression of the biosensor that would not be selected for imaging. We therefore took a single-cell approach, immunostaining the cells for pS473 after stimulation and quantifying the staining intensity in EGFP-positive cells. As shown in high-resolution confocal images in Fig. 1 D, serum-starved cells expressing any EGFP-fusion protein were largely devoid of AKT-pS473 staining. Stimulation of these cells with 10 ng/ml EGF led to the appearance of pS473 staining around the PM (Fig. 1 E). However, cells positive for PH-AKT1-EGFP showed a block in pS473, despite a clear increase in membrane localization of PH-AKT1-EGFP itself (Fig. 1 E). In the non-$PIP_3$–binding PH-AKT1$^{R25C}$-EGFP–positive cells, we still observed an increase in pS473 intensity. In parallel, we imaged cells with a low-resolution 0.75-NA air objective to capture fluorescence from the cells' entire volume, then quantified these images using an automatically determined threshold for GFP-positive cells (see Materials and methods). This analysis showed that the R25C mutant had no substantial effect on pS473 levels, whereas wild-type PH-AKT greatly inhibited pS473 staining in EGF-stimulated cells as well as reducing basal levels in serum-starved cells (Fig. 1 F). Together, these experiments showed that at the population level, PH-AKT1 overexpression has little impact, but at the level of individual cells expressing visible PH-AKT1 (that would be used for imaging experiments), inhibition of AKT activation was profound.

### Genomic tagging of AKT1 as an assay for $PIP_3$ interactions in cells

Whilst inhibition of AKT1 S473 phosphorylation by PH-AKT1 overexpression is consistent with sequestration of that target lipid, it is not a direct demonstration of the phenomenon. We therefore wanted to devise an experiment where we could quantify competition for $PIP_3$ between the biosensor and endogenous AKT1. To that end, we used gene editing to incorporate a bright, photostable neonGreen fluorescent protein to the C terminus of AKT1 via gene editing using a split fluorescent protein approach (Kamiyama et al., 2016). Here, CRISPR/Cas9 is used to cut the 3′ end of the AKT1 open reading frame. Homology-directed repair incorporates the 11th strand of the neonGreen2 protein. This is performed in cells stably expressing the other neonGreen2-1–10 fragment, leading to complementation and fluorescence (Fig. 2 A). Western blotting with AKT1-specific antibodies revealed the presence of the neonGreen2 fusion in edited cells but not parental controls (Fig. 2 B).

Imaging these AKT1-NG2 cells by total internal reflection fluorescence microscopy (TIRFM) revealed diffraction-limited spots at the cell surface, which increased in number with EGF stimulation (Fig. 2 C). We analyzed the intensity of these spots and compared them to intensity distributions from a known monomeric protein localized to the PM and expressed at single-molecule levels, namely a myristoylated and palmitoylated neonGreen protein (produced by fusion to the N terminus of Lyn kinase). Fitting a basis histogram to this known distribution allows the AKT1 data to be fit to a function that will differentiate

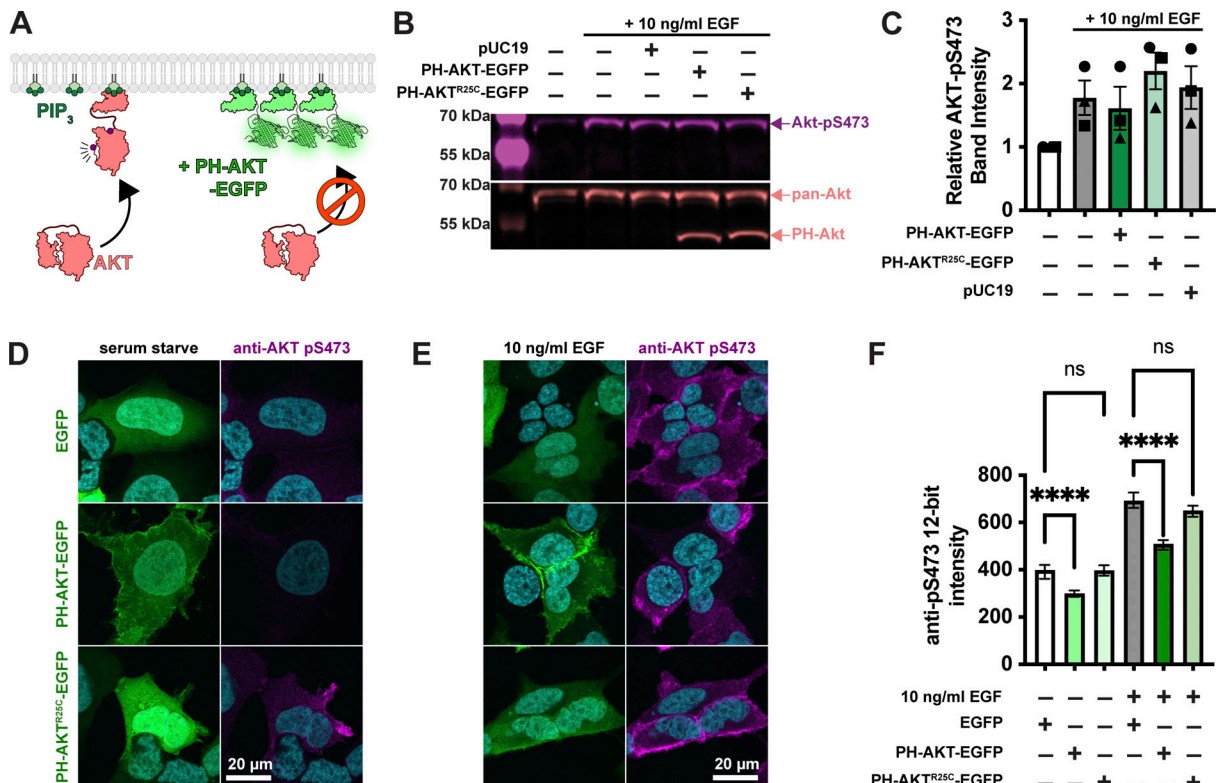

Figure 1. **Single cell analysis reveals profound inhibition of AKT by PH-AKT PIP₃ biosensor. (A)** AKT is autoinhibited by its PIP₃ binding PH domain. PIP₃ production alleviates this steric inhibition, facilitating activating phosphorylations at S473 and T308. Over-expression of PH-AKT is hypothesized to out-compete the endogenous AKT's PH domain. **(B)** Inhibition of AKT S473 phosphorylation is not apparent at the population level. HEK293A cells were serum starved and then treated (where indicated) with 10 ng/ml EGF to activate PI3K. After 5 min, cells were lyzed and analyzed by western blot for total Akt and Akt-pS473. **(C)** Quantification of blots from three experiments like that shown in B. Data are means ± s.e. **(D–F)** Inhibition of Akt activation (via S473 phosphorylation) is apparent through analysis of single cells. HEK293A cells were serum starved (D) or stimulated with EGF (E), then fixed and stained with antibodies against pS473 along with DAPI for nuclear DNA (cyan). **(D and E)** Images show high-resolution confocal micrographs (1.45 NA 100× oil-immersion objective) of representative cells, whereas (F) shows mean fluorescence intensity measurements of cells imaged at low resolution (0.75 NA 20× air objective) to capture fluorescence from the entire volume of the cells. Graphs show medians ± 95% confidence interval of the median from 82 to 160 cells pooled from three experiments (medians are reported since the data are not normally distributed). Results of a multiple comparisons post-hoc test (ns = P > 0.99; **** = P < 0.0001) are indicated from a Kruskal-Wallis test (Kruskall-Wallis statistic = 839.2, six groups, P < 0.0001). Source data are available for this figure: SourceData F1.

between single, dimeric, trimeric, or higher fluorescent molecule intensities in the sample (Mutch et al., 2007), as shown in Fig. 2 D. This revealed that 98.1% (95% confidence interval from six cells = 95.6–100.5%) of the AKT1-NG2 spots were single molecules (Fig. 2 E).

Although we could resolve single molecules of AKT1, video-rate imaging caused significant photobleaching (note the decline in molecule numbers prior to stimulus addition in Fig. 2 C). To circumvent this, we dropped our imaging rate from an image every 50 ms to one every 30 s. Stimulation with 10 ng/ml EGF then revealed a roughly fivefold increase in AKT1 localization at the PM within about 90 s, which then declines with first-order kinetics over the subsequent 10 min of the experiment (Fig. 2 F). This is consistent with the typical evolution of AKT activation in response to growth factor stimulation. Accumulation of AKT-NG2 was ~25 molecules per 100 μm², which assuming a surface area of ~1,500 μm² per cell, corresponds to ~375 molecules total. It should be noted that tagging likely only occurred at a single allele in each cell, and the population still exhibited expression of non-edited

AKT1 (Fig. 2 B). Given that HEK293 are known to be pseudo-triploid (Bylund et al., 2004), the true number of AKT1 molecules would be at least 1,125. However, given an estimated total copy number of 23,000 AKT1 in these cells (Cho et al., 2022), this is still only about 5%. However, we do not interpret these raw numbers due to uncertainties in the efficiency of NG2 complementation under these conditions, as well as potential for reduced expression from the edited allele.

As a further test of PI3K activation, we stimulated PI3K using chemically induced dimerization to recruit the isolated inter-SH2 domain of the PI3K regulatory subunit to the PM. This in turn recruits endogenous, constitutively active p110 catalytic subunits of PI3K, stimulating PIP₃ synthesis (Suh et al., 2006). iSH2 generated a robust and sustained recruitment of AKT1-NG2 molecules to the PM after recruitment of iSH2 (Fig. 2 G). Notably, baseline AKT-NG2 localization increased from ~5 to ~15 per 100 μm² in iSH2 cells, perhaps because the iSH2 construct does not contain the inhibitory SH2 domains of p85 regulatory subunits, producing higher basal PI3K activity.

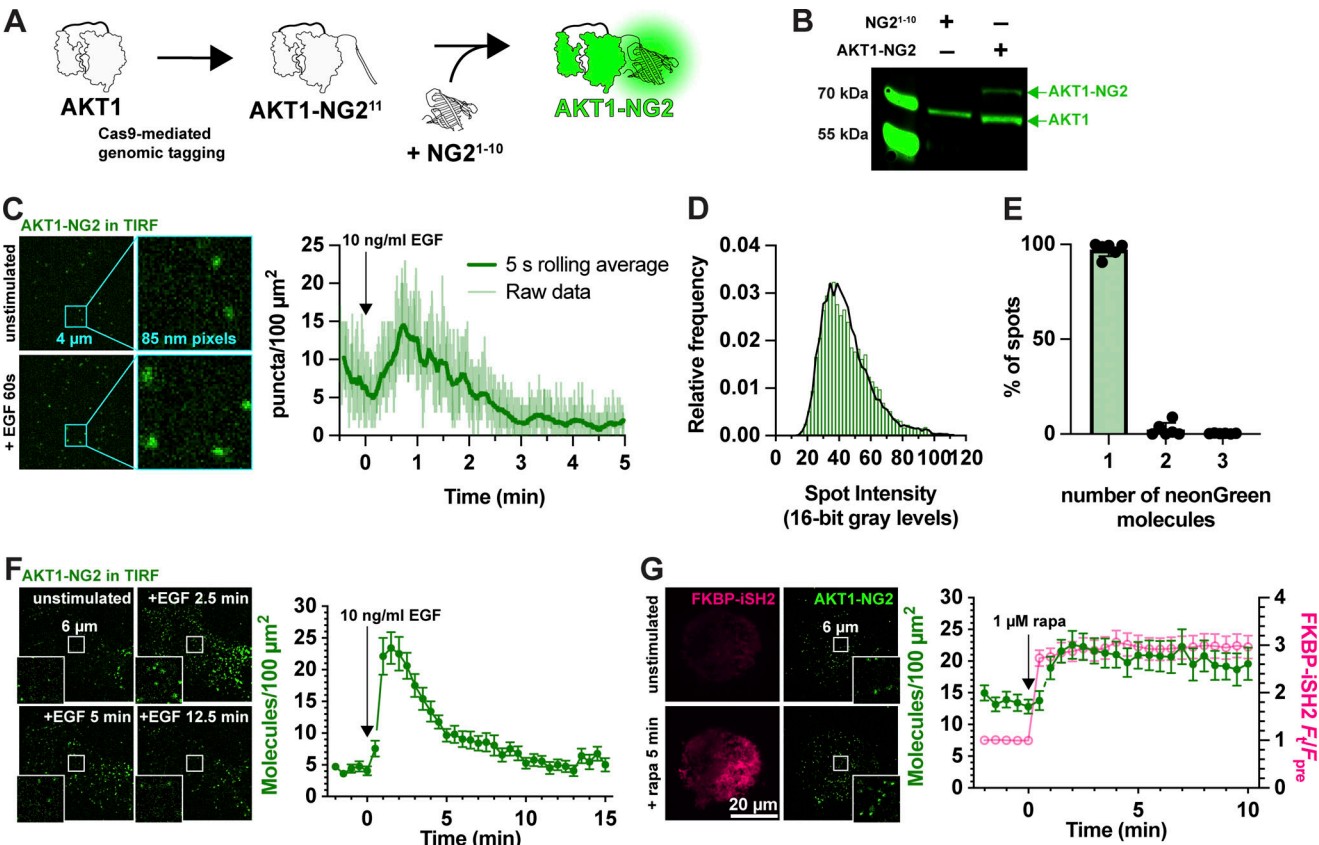

**Figure 2. Fluorescent tagging of AKT1 at its genomic locus. (A)** CRISPR/Cas9 directed cutting at the 3' end of the AKT1 ORF is coupled to homology-directed repair to integrate an in-frame NG2[11] tag, encoding the 11th strand of neonGreen2. The edit is made in a HEK293A cell line stably expressing NG2[1–10], the remainder of the neonGreen protein. The two neonGreen2 protein fragments assemble in the cell to generate fluorescent NG2 protein. **(B)** Western blot of AKT protein from a sorted, polyclonal population of edited cells shows the appearance of a second molecular weight band, consistent with the 81.9 kDa complex between AKT1 (55.7 kDa) and NG2 (26.2 kDa). **(C)** TIRF imaging of AKT1-NG2 cells exhibit discrete fluorescent spots at the cell surface, which increase in density after PI3K activation with 10 ng/ml EGF. The light lines are data points from images recorded at 20 Hz; the thicker, darker line is the 5 s rolling average. **(D and E)** AKT1-NG2 spots are single molecules: (D) The intensity of fluorescent AKT1 spots pooled from a representative experiment shows a mono-modal lognormal distribution (green). The data are fit with a model assuming the intensity is derived from a mixture of monomeric, dimeric or trimeric fluorescent proteins calibrated against a known monomeric mNeonGreen fluorescent protein distribution. The fit predicts 98.1% monomers with a reduced $\chi^2$ of 1.06. **(E)** The results of this analysis pooled across six cells from three independent experiments yields consistent results with mean $\chi^2$ of 1.67 ± 0.40 (s.e.). **(F)** Extended TIRF imaging of AKT-NG2 cells with reduced duty cycle to minimize photobleaching reveals robust recruitment of AKT1 after EGF stimulation. Data are means ± s.e. of 20 cells pooled from two independent experiments. **(G)** PI3K activation is sufficient to recruit AKT1 to the plasma membrane. AKT1-NG2 cells were transfected with PM-targeted Lyn N-terminal 11 residues fused to FRB and the PI3K p110 catalytic subunit-binding iSH2 fused to FKBP and mCherry. 1 μM rapamycin was added to cells to induce dimerization of FRB and FKBP and hence recruitment of iSH2/p110 to the plasma membrane, inducing PIP3 synthesis. AKT1-NG2 is further increased on the PM by this maneuver. Data are means ± s.e. of 32 cells pooled from three independent experiments. Insets in F and G show zoomed view indicated in the center of the cell. Source data are available for this figure: SourceData F2.

### Sequestration of PIP3 by lipid biosensors

Having developed the ability to visualize native AKT1 translocation to the PM in response to PI3K activation, we next designed experiments to test for competition of PIP3 binding by overexpressed PH-AKT1-mCherry biosensor (Fig. 3 A). We transfected AKT1-NG2 cells for 24 h in 35-mm dishes with three different masses of plasmid (all adjusted to 1 μg total plasmid DNA by the addition of inert pUC19 plasmid carrier). When imaged at the single-cell level in TIRFM, this led to a somewhat weak correlation of transfected plasmid mass with single-cell mCherry fluorescence intensity (Fig. 3 B). Nonetheless, when these cells were stimulated with EGF, cells transfected with any dose of PH-AKT1 showed robust recruitment of the biosensor to the PM (Fig. 3, C and D). However, at all three doses, we

observed complete ablation of endogenous AKT1-NG2 translocation to the PM (Fig. 3, C and E). Thus, the presence of the biosensor seemed to prevent engagement of endogenous AKT1 with PIP3. To observe the concentration dependence of AKT1-PH-mCherry inhibition, we pooled the single-cell data from these experiments and split transfected cells into cohorts based on raw expression level (excitation and gain were consistent between experiments, allowing direct comparison). This analysis showed profound inhibition of AKT1-NG2 recruitment at all expression levels, with a slightly reduced effect only visible in the lowest expressing cohort (Fig. 3 F).

Prior studies have reported inhibition of AKT signaling by the AKT1 PH domain, though this was attributed to secondary interactions of the PH domain with undefined effectors; it was

Figure 3. **PH-AKT1 PIP₃ biosensor abolishes endogenous AKT recruitment to the PM. (A)** Hypothesized competition for PIP₃ between the overexpressed PH domain and endogenous AKT1-NG2. **(B)** Raw 16-bit intensity levels of mCherry fluorescence in HEK293A cells transiently transfected with the indicated mass of PH-AKT1-mCherry for 24 h. Small data points represent measurements of individual cells, whereas large points are the means of each of three independent experiments. Points are color matched by experiment. Grand means ± s.e. are also indicated. **(C)** TIRF imaging of AKT1-NG2 cells from (B) stimulated with 10 ng/ml EGF. Insets show zoomed view indicated in the center of the cell. **(D)** Relative PM fluorescence of PH-AKT1-mCherry during time lapse imaging and stimulation with 10 ng/ml EGF. **(E)** As in D, except the density of endogenous AKT1-NG2 molecules are counted. Data in D and E are grand means of three experiments ± s.e. imaging 8–10 cell each. **(F)** Data from E were pooled from all three experiments, and raw 12-bit mCherry fluorescence intensity data (constant laser power and camera gain between experiments) was used to bin cellular measurements as indicated on the x-axis. Small points represent individual cell measurements, whereas the large symbols are means ± 95% confidence interval. Data were normal by Kolmogorov-Smirnov test, and results of a Holm-Šídák's multiple comparisons test are indicated comparing to control cells ("0" cohort); *** P = 0.002, **** P < 0.0001, performed post-hoc to an ordinary one-way ANOVA (F = 36.73, P < 0.0001).

not observed with other PIP₃ biosensors (Várnai et al., 2005). We therefore tested additional PIP₃ biosensors (Fig. 4 A). We examined the commonly used BTK PH domain (Várnai et al., 1999), along with the 2G splice variant of the ARNO PH domain (Venkateswarlu et al., 1998), carrying an I303E point mutation to disrupt interactions with Arl-family GTPases (Goulden et al., 2019). This latter construct was expressed as a high-avidity tandem dimer (aPHx2) or else as a low-copy monomer (aPHx1) expressed from a truncated CMV promoter (Morita et al., 2012). Both PH-BTK and aPHx2 completely blocked AKT1-NG2 translocation, whereas the lower affinity aPHx1 construct reduced AKT1-NG2 translocation by around 50% (Fig. 4 B), which was highly significant (Fig. 4 C). Therefore, sequestration of PIP₃ occurred with a variety of PIP₃ biosensors.

Although PIP₃ is the principle activator of AKT, the enzyme can also be activated by the PIP₃ degradation product, PI(3,4)P₂ (Frech et al., 1997; Ebner et al., 2017). We therefore tested whether PI(3,4)P₂ biosensors could disrupt AKT1-NG2 translocation (Fig. 4 D). A high-avidity tandem trimer of the TAPP1 PH domain, cPHx3, is a sensitive reporter for PI(3,4)P₂ production (Goulden et al., 2019), and this construct produced a notable inhibition (but not ablation) of AKT1-NG2 recruitment (Fig. 4 D). On the other hand, a single PH domain version of the same sensor, cPHx1, expressed from a truncated CMV promoter was without significant effect (Fig. 4, E and F). Therefore, PIP₃ biosensors produced a more profound inhibition than PI(3,4)P₂ biosensors. This is not surprising, since complete PI(3,4)P₂

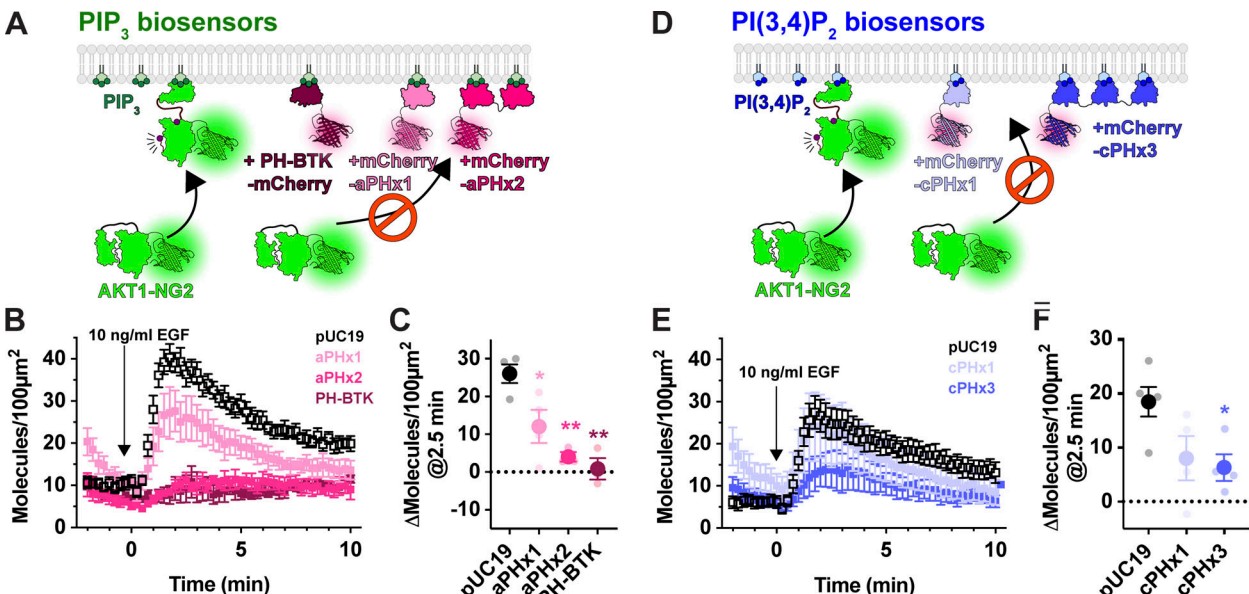

**Figure 4. PIP$_3$ and PI(3,4)P$_2$ biosensors inhibit endogenous AKT recruitment to the PM. (A)** Hypothesized competition for PIP$_3$ between PIP$_3$ biosensors and endogenous AKT1-NG2. **(B)** AKT-NG2 cells were transiently transfected with the indicated PIP$_3$ biosensors, or pUC19 as an inert control, and imaged after 24 h by TIRFM. Data quantify AKT-NG2 density at the cell surface. The time-lapse data are grand means ± s.e. of 3–4 independent experiments imaging 9–10 cells each. **(C)** The scatter plots show individual experiment means (small symbols) and grand means ± s.e. **(D)** As in A, except high and low avidity versions of a PI(3,4)P$_2$-selective biosensor were tested. **(E)** Data are grand means ± s.e. of 3–5 independent experiments imaging 6–10 cells each. **(F)** summary data for E as described in C. Results of a Tukey's multiple comparisons test are indicated comparing to pUC19 cells * P < 0.05, ** P < 0.005, performed post-hoc to an ordinary one-way ANOVA (C: F = 12.55, P = 0.0010; F: F =4.702, P = 0.0363).

sequestration would not affect the ability of AKT1-NG2 to engage PIP$_3$, whereas PIP$_3$ biosensors might also reduce the ability of 5-OH phosphatases to convert PIP$_3$ to PI(3,4)P$_2$ (Trésaugues et al., 2014).

### Mitigating PIP$_3$ competition using biosensors expressed at single-molecule levels

The sequestration of native PIP$_3$ by overexpressed biosensors is not surprising when comparing the fluorescence intensity of endogenously labelled AKT1 with the biosensors. AKT1-NG2 expressed from a native allele produces fluorescence that is resolved as single molecules (Fig. 2, D and E) and achieves densities of 25–35 molecules per 100 µm$^2$. On the other hand, PH-AKT1-mCherry produces an intense, even labelling of the PM when viewed in TIRFM (e.g., Fig. 3 C). This profile is produced by the high densities of biosensor molecules, which cannot be resolved at the diffraction limit and instead convolve their fluorescence into a monolithic haze. Given an estimated Airy disc of an mCherry molecule of 210-nm diameter, or 0.035 µm$^2$ through our 1.45 NA optics, a minimum density to achieve such convolution would be around 3,000 molecules per 100 µm$^2$. Therefore, the biosensor molecules clearly outnumber the endogenous effectors by at least two orders of magnitude. We then reasoned that if we dropped biosensor expression levels to be comparable with native AKT1, sequestration of PIP$_3$ might be prevented (Fig. 5 A).

To accomplish this goal, we expressed aPHx1 and cPHx1 PIP$_3$ and PI(3,4)P$_2$ biosensors tagged with iRFP670 from truncated CMV promoters (Morita et al., 2012) with <4 h between

transfection and imaging. Under these conditions, we selected cells with small numbers of individual fluorescent puncta with intensities consistent with single molecules (Fig. 5 B). Typically, these cells exhibited 5–20 molecules per 100 µm$^2$ of PM prior to stimulation. Crucially, in AKT1-NG2 cells expressing these biosensors, we saw no inhibition of AKT1 translocation after EGF stimulation (Fig. 5 C). In these experiments, both aPHx1 and cPHx1 showed robust translocation to the membrane, similar to AKT1-NG2 (Fig. 5 D). Normalizing the data to the maximum response revealed a rapid onset of the signal followed by a decay over about 10 min (Fig. 5 E). The recruitment also revealed the lagging kinetics of the PI(3,4)P$_2$ reporter, owing to the synthesis of this lipid from PIP$_3$ (Hawkins et al., 1992).

The fact that PIP$_3$ and PI(3,4)P$_2$ biosensors expressed at single-molecule levels do not inhibit PI3K signaling justifies a use case for this approach. However, comparing performance with their overexpressed counterparts revealed additional benefits. Normalizing the aPHx1 PIP$_3$ sensor count to fold increase over baseline allows comparison with the strongly expressed aPHx1 data (from the experiments introduced in Fig. 4): this showed that when expressed at single-molecule levels, the biosensor has substantially better dynamic range, with a fold increase over baseline of nearly 100%, compared with 60% for the same probe overexpressed (Fig. 5 F). Normalizing data from both expression modes to their maximum response (Fig. 5 G) and fitting kinetic profiles for cooperative synthesis and degradation reactions revealed the rate of synthesis is remarkably similar: 1.09 min$^{-1}$ (95% C.I. 1.02–1.17) for single-molecule expression versus 1.02 min$^{-1}$ (95% C.I. 0.98–1.06) for overexpression. On

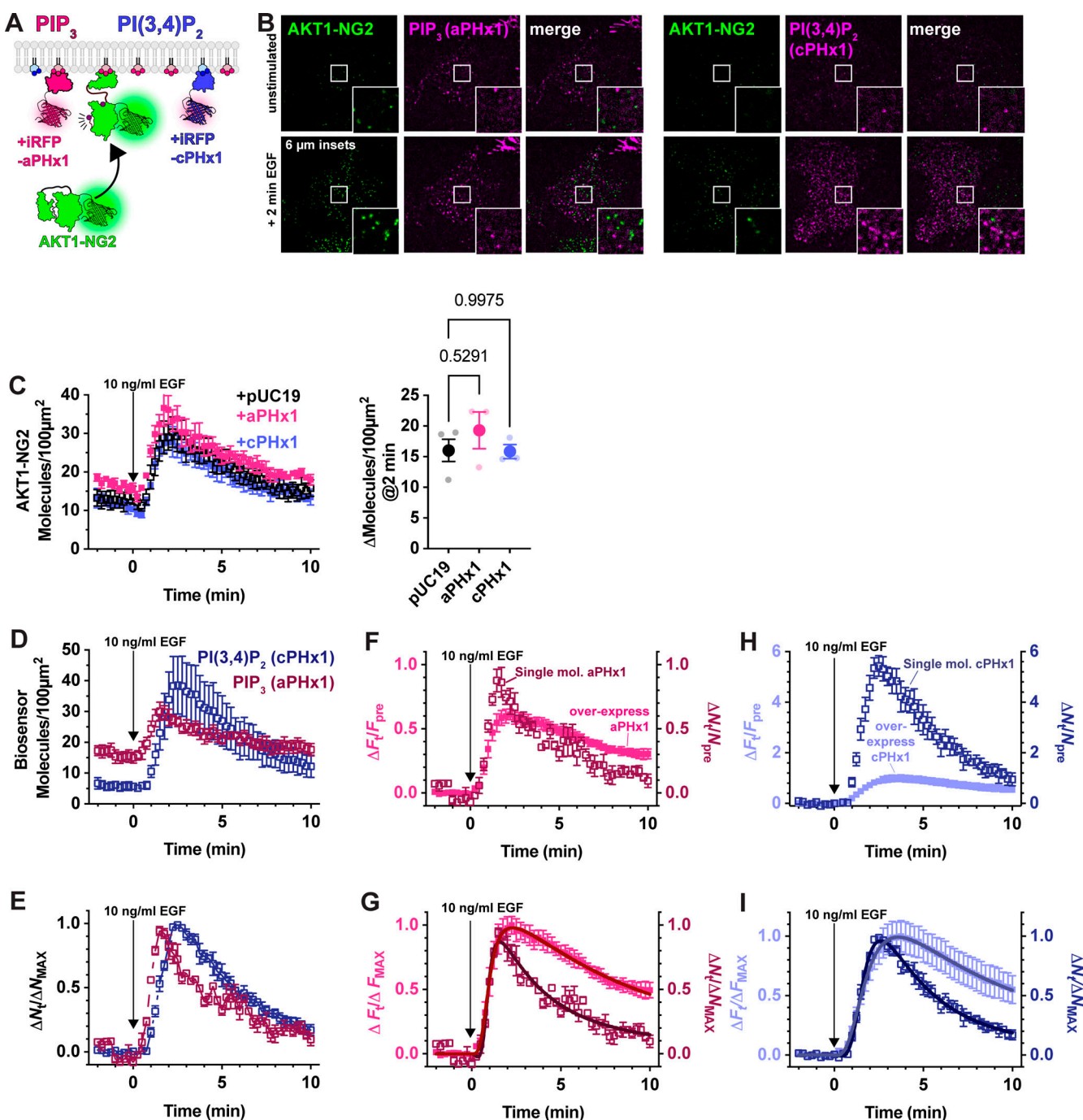

Figure 5. **Weak expression of PIP$_3$ and PI(3,4)P$_2$ biosensors does not inhibit AKT1 translocation, and reveals improved dynamic range and kinetic fidelity. (A)** Hypothesized weak expression of biosensors that do not sequester a large fraction of the available lipids. **(B)** Example TIRFM images of AKT1-NG2 cells transiently transfected with aPHx1 PIP$_3$ or cPHx1 PI(3,4)P$_2$ probes for 2–4 h before imaging and stimulation with EGF. Note the single molecule expression of the biosensors. Insets show zoomed view indicated in the center of the cell. **(C)** Quantification of AKT1-NG2 translocation of aPHx1, cPHx1 or pUC19 (control)-transfected cells, showing no effect. The scatter plot shows individual experiment means (small symbols) and grand means ± s.e. P values of a Tukey's multiple comparisons test are indicated comparing to pUC19 cells, performed post-hoc to an ordinary one-way ANOVA (F = 0.8206, P = 0.4784). **(D)** Quantification of the lipid biosensor density at the cell surface from the same experiments. **(E)** As in D, except data were normalized to the maximum density for each experiment, showing the characteristic lagging accumulation of PI(3,4)P$_2$ versus PIP$_3$. **(F–I)** Comparison of weakly (single-molecule level) expressed biosensors with the same constructs strongly over-expressed (24 h). **(F)** comparison of the change in fluorescence or molecule density versus baseline for weakly and strongly expressed PIP$_3$ biosensor, aPHx1. **(G)** as in F, but data are normalized to the maximum for each experiment. **(H)** comparison of the change in fluorescence or molecule density versus baseline for weakly and strongly expressed PI(3,4)P$_2$ biosensor, cPHx1. **(I)** As in I, but data are normalized to the maximum for each experiment. Data are grand means ± s.e. of 3–4 experiments analyzing 9–10 cells each. The data for the over-expressed biosensor were generated from the same experiment as presented in Fig. 4. Solid lines in G and I show fits to two co-operative reactions (synthesis and degradation). See text for details.

the other hand, degradation slowed with over expression from 0.34 min$^{-1}$ (95% C.I. 0.24–0.58) to 0.13 min$^{-1}$ (95% C.I. 0.12–0.15). This is expected, since synthesis of PIP$_3$ molecules would not be prevented by biosensor. On the other hand, PIP$_3$ degradation could be slowed by the overexpressed biosensor competing with PTEN and 5-OH phosphatases that degrade PIP$_3$. An even more exaggerated result is achieved with the cPHx1 PI(3,4)P$_2$ biosensor; this shows an increase in fold change over baseline of 600% for single-molecule expression levels compared with only 100% in overexpressed cells (Fig. 5 H). Again, the degradation rate of the signal is substantially slowed by the overexpressed sensor, reducing from 0.27 min$^{-1}$ (95% C.I. 0.22–0.39) to 0.16 min$^{-1}$ (95% C.I. 0.14–0.19), whereas synthesis remains only minorly impacted, changing from 0.61 min$^{-1}$ (95% C.I. 0.57–0.64) to 0.54 min$^{-1}$ (95% C.I. 0.52–0.56) with overexpression (Fig. 5 I). Collectively, these data show that single molecule–based PI3K biosensors show improved dynamic range and kinetic fidelity compared with the same sensors overexpressed.

## Discussion

Inhibition of lipid effector protein function by lipid biosensors is one of the most commonly raised concerns with these probes (Balla et al., 2000; Maekawa and Fairn, 2014; Wills et al., 2018). Indeed, overexpression of lipid-binding domains has been shown to inhibit lipid functions of PI3P, PI(4,5)P$_2$, and PIP$_3$ (Simonsen et al., 1998; Balla and Várnai, 2002; Várnai et al., 2005). What is not clear is whether this is due to sequestration of the lipid itself. An alternative is competitive binding of a tertiary co-interactor by the lipid:biosensor complex, as for example observed between cytohesin family PH domains, PIP$_3$, and Arf family small GTPases (Cohen et al., 2007; Li et al., 2007; Hofmann et al., 2007). Notably, competitive binding of a co-interactor would not necessarily prevent lipid-dependent localization of an endogenous lipid effector protein, whereas sequestration of the lipid itself would. Here, we unambiguously demonstrate direct competitive binding of PIP$_3$ by lipid biosensors, preventing membrane localization of a key effector protein, AKT1 (Figs. 3 and 4), and leading to inhibition of AKT activation in cells (Fig. 1). Notably, we observed PIP$_3$ sequestration by three different PIP$_3$-binding modules, again arguing against co-interactors being critical for the inhibition.

How general a problem is lipid sequestration by biosensors likely to be? As we have argued here and elsewhere (Wills et al., 2018), this will depend on the bulk levels of the lipids and their effector proteins. In support of this, we observed that dropping biosensor expression to low levels matching that of AKT1 produced no inhibition of AKT1 recruitment Fig. 5). Therefore, standard techniques of transient transfection for strong expression of biosensors are likely only an issue for molecules present at exceptionally low levels, i.e., transiently generated signaling molecules such as PIP$_3$, and perhaps other scarce phosphoinositides such as PI5P and PI(3,5)P$_2$.

Intriguingly, we report a counterintuitive finding that weaker expression of the PI3K lipid biosensors actually produces increased sensitivity of the probes (Fig. 5). This is almost certainly

due to the failure of the lipids to become saturated by biosensor molecules (which causes endogenous effectors to be outcompeted). We also observed enhanced kinetic fidelity, perhaps because overexpressed biosensors prevent access of PIP$_3$-degrading phosphatases or effectors that stimulate negative feedback. Either way, this single-molecule approach will permit precise and quantitative analysis of PI3K pathway activation in living cells with greater accuracy.

## Materials and methods

### Cell culture, transfection, and gene editing

HEK293A cells (R70507; Thermo Fisher Scientific) were cultured in low glucose DMEM (10567022; Thermo Fisher Scientific) supplemented with 10% heat-inactivated fetal bovine serum (10438-034; Thermo Fisher Scientific), 100 µg/ml streptomycin + 100 U/ml penicillin (15140122; Thermo Fisher Scientific), and 0.1% (vol/vol) chemically defined lipid supplement (11905031; Thermo Fisher Scientific) at 37°C and 10% atmospheric CO$_2$ in humidified incubators. Passaging was performed 1:5 by rinsing with PBS and dissociating the cells in TrpLE (12604039; Thermo Fisher Scientific) and diluting 1:5 in fresh media.

Gene editing was performed following previously published protocols (Kamiyama et al., 2016; Zewe et al., 2018). We used the strategy to combine CRISPR/Cas9–mediated targeting of an allele with homology-directed repair to tag AKT1 at the C terminus with a split neonGreen variant (NG2-11) in an HEK293A cell line stably overexpressing the remainder of the neonGreen protein (NG2-1-10), as described by the OpenCell project (Cho et al., 2022). The guide RNA protospacer sequence was 5'-AGCGGCACGGCCTGAGGCGG-3' (ordered as ThermoFisher custom gRNA) and the homology-directed repair template synthesized as a single-stranded "ultramer" primer sequence (IDT) 5'-CAGCGAGCGCAGGCCCCACTTCCCCCAGTTCTCCTACTCGGCCAGCGGCACGGCCGGTGGCGGATTGGAAGTTTTGTTTCAAGGTCCAGGAAGTGGTACCGAGCTCAACTTCAAGGAGTGGCAAAAGGCCTTTACCGATATGATGTGAGGCGGCGGTGGACTGCGCTGGACGATAGCTTGGAGGGATGGAGAGGCGGCCT-3'. A Neon electroporation system (Thermo Fisher Scientific) was used to introduce the components according to the manufacturer's instructions. Essentially, 10 pmol of gRNA was preincubated with 10 pmol TruCUT Cas9 protein V2 in 5 µl buffer R for 20 min before adding 100 pmol of the homology-directed repair template. The mixture was then added to 200,000 cells in 5 µl buffer R and electroporated with a single 20-ms, 1,500-V pulse. Cells were seeded in complete antibiotic-free media in 6-well plates and left to recover for 48 h before screening for fluorescence by confocal microscopy. After estimating the fraction of edited cells at ~1%, the positive cells were sorted by FACS and expanded, resulting in a polyclonal population.

HEK293A cells were seeded in either 6-well plates for immunofluorescence and western blotting experiments or 35-mm dishes containing 20-mm #1.5 optical glass bottoms (CellVis D35-20-1.5-N) for live-cell imaging. Dishes were pre-coated with 10 µg ECL cell attachment matrix (08-110; Sigma-Aldrich) for 60 min. Transfection used Lipofectamine 2000 (11668019; Thermo Fisher Scientific) according to the manufacturer's instructions.

Briefly, 1 µg total DNA was precomplexed with 3 µg Lipofectamine 2000 in 0.2 ml Opti-MEM (51985091; Thermo Fisher Scientific) for ≥5 min before adding to the cells in 2 ml media. Plasmids used for transfection are listed in Table 1. After 4 h, media was replaced to remove the transfection reagent. 24 h prior to stimulation with EGF, cells were serum starved in Fluorobrite (A1896702; Thermo Fisher Scientific) supplemented with 0.1% chemically defined lipid supplement (11905031; Thermo Fisher Scientific) and 0.1% BSA.

## Western blotting

Cells in 6-well plates were treated as described in figure legends before lysis in 150 µl ice-cold RIPA buffer, including protease (539131-1VL; Sigma Millipore) and phosphatase (524627; Sigma Millipore) inhibitor cocktails. After scraping and clearing at 10,000 $g$, 40 µl lysates were boiled for 5 min in Bolt LDS Sample Buffer (B0008; Thermo Fisher Scientific) before running on 12% Bis-Tris gels at 165 V, then transferred to nitrocellulose membrane (LC2000; Novex) at 10 V. After transfer, the membrane was blocked in Tris-buffered saline (50 mM Tris with 150 mM NaCl) with 0.05% Tween-20 containing 1% nonfat dry milk (9999S; Cell Signaling Technology) for 1 h. Blots were stained with primary antibodies pan AKT clone 40D4 (2920; Cell Signaling Technology), Phospho-AKT (Ser473) monoclonal antibody (4060; Cell Signaling Technology ), Akt1 (2938; Cell Signaling Technology), or DM1A (62204; Thermo Fisher Scientific) and secondary antibodies Alexa647 goat anti-rabbit (A-21245; Thermo Fisher Scientific), Alexa555 goat anti-mouse IgG1 secondary antibody (A-21127; Thermo Fisher Scientific), or Alexa800 goat anti-mouse IgG (A32730; Thermo Fisher Scientific).

## Immunofluorescence

4 h after transfection, cells were dissociated and reseeded at 25% confluence in 40 µl onto 8-well multi-test slides (Electron Microscopy Sciences Hydrophobic PTFE Printed Microscope Slide 8 Round Wells 6 mm, 6342206) pre-coated with 0.8 µg ECL and returned to the incubator for 24 h. Cells were serum starved 1.5 h before stimulation with Fluorobrite supplemented with 0.1% chemically defined lipid supplement and 0.1% BSA, before stimulating where indicated with 10 ng/ml EGF (354052; Corning). After 5 min of stimulation, cells were fixed by the addition of formaldehyde (Electron Microscopy Sciences 16% Paraformaldehyde Aqueous Solution, EM Grade, Ampoule 10 ML 15710) in PBS to 4% final concentration for 15 min at room temperature. Cells were rinsed three times in 50 mM NH$_4$Cl in PBS before blocking and permeabilization in blocking solution (5% normal goat serum and 0.2% Triton X-100 in PBS) for 30 min. Cells were then stained in anti-PhosphoAKT (Ser473), followed by Alexa647 goat anti-rabbit secondary, GFP-booster-Atto488 (Chromotek gba-488), DAPI, and HCS CellMask Orange (H32713; Thermo Fisher Scientific). Cells were rinsed with PBS then Millipore water, then mounted in ProLong Diamond (P36961; Thermo Fisher Scientific) before imaging on a Nikon A1R confocal microscope attached to a Nikon TiE inverted stand. Imaging used a 20× plan apochromatic 0.75 NA air immersion objective with the confocal pinhole fully open for quantification. For some experiments, we also captured high-resolution confocal

Table 1.  **Plasmids used in this study**

| Plasmid | Backbone | Insert | Reference |
|---|---|---|---|
| EGFP | pEGFP-N1 | None | |
| PH-AKT1-EGFP | pEGFP-N1 | *AKT1*(1–164):DPPVAT:*EGFP* | |
| PH-AKT1$^{R25C}$-EGFP | pEGFP-N1 | *AKT1*(1–164)-*R25C*:DPPVAT:*EGFP* | |
| PH-PLCD1-mNeonGreen | pmNeonGreen-N1 | *PLCD1*(1–170):GVGG:*mNeonGreen* | |
| Lyn$^{N11}$-FRB-iRFP | piRFP713-N1 | *LYN*(1–11):*MTOR*(2021–2113):*iRFP713* | |
| Lyn$^{N11}$-mNeonGreenx1 | pNeonGreen-N1 | *LYN*(1–11):GVGG:*mNeonGreen* | |
| Lyn$^{N11}$-mNeonGreenx2 | | *LYN*(1–11):GVGG:*mNeonGreen*:SPVAT:*mNeonGreen* | |
| Lyn$^{N11}$-mNeonGreenx3 | | *LYN*(1–11):GVGG:*mNeonGreen*:SPVAT:*mNeonGreen*:SPVAT:*mNeonGreen* | |
| mCherry-FKBP-iSH2 | | *mCherry*: SGLRSRAALG:*FKBP1A*(3–108): SA[GGSA]$_4$PRAQAS:*Mus musculusPIK3R2*(420–615) | Suh et al. (2006) |
| PH-AKT1-mCherry | pmCherry-N1 | *AKT1*(1–164):DPPVAT:*mCherry* | |
| PH-BTK-mCherry | pmCherry-N1 | *BTK*(1–177):DPPVAT:*mCherry* | Várnai et al. (1999) |
| NES-iRFP670-aPHx1 | pCMVd3-iRFP670-C1$^a$ | *X. leavis map2k1.L*(32–44):PVAT:*iRFP670*:SGLRSRAQASNSAVDM:*CYTH2i2*(252–399)-*I303E* | Goulden et al. (2019) |
| NES-mCherry-aPHx2 | pCMVd3-mCherry-C1$^a$ | *X. leavis map2k1.L*(32–44):PVAT:*mCherry*:SGLRSRAQASNSAVDM:*CYTH2i2*(252–399)-*I303E*:GGGGGATCGGGTGGTGTCGACATG:*CYTH2i2*(252–399)-*I303E* | Goulden et al. (2019) |
| NES-iRFP670-cPHx1 | pCMVd3-iRFP670-C1$^a$ | *X. leavis map2k1.L*(32–44):PVAT:*iRFP670*:SGLRSRAGGAGAILS:*PLEKHA1*(169–329) | Goulden et al. (2019) |
| NES-mCherry-aPHx3 | pCMVd3-mCherry-C1$^a$ | *X. leavis map2k1.L*(32–44):PVAT:*mCherry*:SGLRSRAQASNSTWKMSS:*PLEKHA1*(169–329):GGSGGSGG:*PLEKHA1*(169–329):GGSGGSGG:*PLEKHA1*(169–329) | Goulden et al. (2019) |

All genes (except fluorescent proteins) are human unless otherwise stated. Amino acid linkers are indicated with single letter code.
$^a$pCMVd3 plasmids are based on the pEGFP-C1 backbone, with the CMV promoter truncated to remove 18 of the 26 putative transcription factor–binding sites in the human cytomegalovirus major intermediate enhancer/promoter (pCMVΔ3 as described in Morita et al. [2012]).

images (pinhole set to 1 Airy unit) using a 100× 1.45 NA oil-immersion objective. Excitation was with 405, 488, 561, or 647 nm on a fiber-coupled 4-line excitation LU-NV laser combiner for DAPI, Atto488/EGFP, CellMask, and Alexa647, respectively. Emission was collected using 425–475-nm (DAPI), 500–550-nm (EGFP), or 663–737-nm (Alexa 647) band passes for each channel on separate line scans. Identical laser excitation power, scan speeds, and photomultiplier gains were used across experiments to enable direct comparison.

### Live-cell imaging
Imaging was performed on a Nikon TiE inverted microscope stand with motorized TIRF illuminator (Nikon) fiber-coupled to a four line Oxxius laser launch equipped with 405-, 488-, 561-, and 638-nm laser lines. A 100× 1.45 NA plan apochromatic oil-immersion objective was used, combined with a 1.5× magnifier. Images were collected on a Hamamatsu Fusion-BT sCMOS camera in ultra-low noise mode with 2× pixel binning. mCherry (561-nm excitation) fluorescence was collected through a dual pass 420–480- and 570–620-nm filter (Chroma), neonGreen and EGFP (488-nm excitation) used a 500–550-nm bandpass filter (Chroma), and iRFP680 (638-nm excitation) used a dual pass 505–550- and 650–850-nm filter (Chroma).

### Image analysis
All image analysis was performed using the open access ImageJ implementation, Fiji (Schindelin et al., 2012). For immunofluorescence, we identified individual cells by auto thresholding the DAPI channel using the "Huang" method, followed by the Watershed function to segment bunched cells that appeared to touch. We then used the Voronoi function to generate boundary lines for the segmentation of the cells. To identify cytoplasm, auto thresholding of the CellMask channel using the "Huang" function was employed, with the cells segmented by adding the nuclear Voronoi boundaries. The "analyze particles" function was then used to identify individual cellular ROIs that were >10 μm² and were not touching the image periphery. These ROIs were used to measure the raw 12-bit intensity of the EGFP and AKT-pS473 channels. A cutoff of EGFP >100 was used to define EGFP-positive cells, since this value was greater than the mean ± 3 standard deviations of the non-transfected cells' EGFP intensity. Background intensity of AKT-pS473 was estimated from control cells subject to immunofluorescence in the absence of AKT-pS473 antibody; this value was subtracted from the measured values of all other conditions.

For analysis of overexpressed biosensor intensity, background intensity was subtracted, defined using the modal pixel intensity that corresponded to the pixel intensity of the camera sensor outside of cells. Manually drawn ROI were then used to measure the raw fluorescence intensity over time, which was normalized to the pre-stimulation fluorescence intensity value. For counting single fluorescent molecules, we employed a macro that took a manually defined 100 μm² ROI for each cell footprint and employed the Fiji plugin ThunderSTORM (Ovesný et al., 2014) to count single-molecule localizations. After entering the pixel and camera calibrations into this plugin, we used the default molecule detection parameters. Data were plotted in GraphPad Prism 9 or later.

### Fitting of reaction kinetics
Curve fitting was performed in GraphPad Prism 9 or later. For the data presented in Fig. 5, G and I, both synthesis and degradation phases displayed clear "s" shaped profiles not well fit by simple first-order kinetics. Since activation of the PI3K pathway involves many multiplicative interactions between adapters and allosteric activation of the enzymes themselves, we assumed cooperativity and fit reactions with the two-phase reaction as follows:

$$F_t = \begin{cases} F_0, \wedge t < 0 \\ F_{max}\left(\dfrac{t^{n_{syn}}}{t^{n_{syn}} + \tau_{syn}^{n_{syn}}} - \dfrac{t^{n_{deg}}}{t^{n_{deg}} + \tau_{deg}^{n_{deg}}}\right), \wedge t \geq 0 \end{cases}.$$

Here, $F_t$ denotes $\Delta F_t / \Delta F_{MAX}$, $n_{syn}$ and $n_{deg}$ are the Hill coefficients of the respective synthesis and degradation reactions, and the rate constants for the reactions are derived from $k_{syn} = 1/\tau_{syn}$ and $k_{deg} = 1/\tau_{deg}$.

### Analysis of single-molecule intensities
Polydispersity of fluorescent puncta was calculated by deconvolution of intensity distributions (Mutch et al., 2007). The intensity distribution of a population of N fluorescent puncta of unknown polydispersity can be expressed as

$$\rho(x) = \frac{1}{N}\sum_{c=1}^{M} A_c \rho_c(x),$$

where $A_c$ denotes the number of puncta containing c fluorophores, $\rho c(x)$ denotes the mean intensity distribution of a population of puncta with c fluorophores, and M is the highest number of fluorophores to be found in a single punctum.

The c = 1 basis histogram was generated from intensity distributions acquired from in vivo single-molecule tracking measurements of a known monomeric reporter (Lyn^N11-mNeonGreenx1). Higher order basis histograms were calculated through a transformation of the c = 1 histogram by

$$\rho_c(x) = \rho_1(x/c)d(x/c).$$

Multimer population proportions were calculated by reduced chi-squared fitting of randomly generated intensity distributions derived from basis histograms to experimental intensity distributions.

### Online supplemental material
Data S1 shows source data for Figs. 1, C and F; 2, C–F; 3, B and D–F; 4, B, C, E, and F; and 5, C–H. Also available at https://doi.org/10.5281/zenodo.14617411.

### Data availability
Source data for Fig. 1, B, C, and F; Fig. 2, B–F; Fig. 3, B–F; Fig. 4, B, C, E, and F; and Fig. 5, C–H are available at https://doi.org/10.5281/zenodo.14617411.

## Acknowledgments
We are grateful to Dr. Tamas Balla (National Institutes of Health, Bethesda, MD, USA) for the kind gift of PH-AKT1-EGFP,

PH-AKT1$^{R25C}$-EGFP, and PH-BTK-EGFP plasmids. We thank two anonymous peer reviewers and André Nadler for their extremely helpful comments on the manuscript.

This work was supported by the National Institute of General Medical Sciences award number 2R35GM119412.

Author contributions: V.L. Holmes: investigation and writing—review and editing. M.M.C. Ricci: conceptualization, data curation, formal analysis, investigation, methodology, visualization, and writing—original draft, review, and editing. C.C. Weckerly: investigation, methodology, resources, and writing—review and editing. M. Worcester: formal analysis and software. G.R.V. Hammond: conceptualization, data curation, formal analysis, funding acquisition, project administration, resources, software, supervision, and writing—original draft, review, and editing.

Disclosures: The authors declare no competing interests exist.

Submitted: 3 December 2024

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
