## [Peer Review File · The Journal of Cell Biology]

Single molecule Lipid Biosensors Mitigate Inhibition of Endogenous Effector Proteins

Victoria Holmes, Morgan Ricci, Claire Weckerly, Michael Worcester, and Gerald Hammond

Corresponding Author(s): Gerald Hammond, University of Pittsburgh School of Medicine

Review Timeline:

Submission Date:	2024-12-03
Editorial Decision:	2025-01-07
Revision Received:	2025-01-07

Monitoring Editor: Lois Weisman

Scientific Editor: Andrea Marat

Transaction Report:

DOI: <https://doi.org/10.1083/jcb.202412026>

Revision 0

Review #1

1. Evidence, reproducibility and clarity:

Evidence, reproducibility and clarity (Required)

The authors present the use of previously identified biosensors in a single-molecule concentration regime to address lipid effector recruitment. Using controlled and careful single-cell based analysis, the study investigates how expression of the commonly used PIP3 sensor based on Akt-PH domain interferes with the native detection of PIP3. Predominantly live-cell fluorescence microscopy coupled to image analysis drives their studies.

Conceptually, this manuscript carefully and quantitatively describes the influence of lipid biosensor overexpression and presents a means to overcome the inherent and long-recognized problems therein. This solution, namely employing low expression of the lipid biosensor, should be generally applicable. The work is of general interest to cell biologists focused on answering questions at membranes and organelles, including especially those interested in lipid-mediated signaling transductions.

****Major:****

1. The terminology "single molecule biosensor" is not really appropriate. A protein is not "single-molecule". An enzyme does not "single molecule". Better is biosensors at single-molecule expression levels. In most cases, this should be changed. Single-molecule vs single-cell vs. bulk measurements are often poorly defined in quantifications and conflating these does not help the case, which is already supported by generally clear data.
2. Figure 1D-F, images not as clearly describing quantitation as one would hope. Untransfected cells in 1E should demonstrate more translocated Akt-pS473 than transfected, but it is difficult for this reviewer to find. Consider inset images in addition to the wider field. Consider also moving the "negative" data of Fig 1B-C to Supplement.
3. The cell line being used is not clearly specified after the initial development of the NG1 followed by CRISPRed NG2 onto Akt. For example, for the Figure 3C experiments, the text states "complete ablation of endogenous AKT1-NG2" but this information is not apparent from the figure legend or figure. Throughout the cell line used and the aspects transfected need to be made explicitly clear.
4. Fig. 5 shows single cells. It is therefore unclear if broken promoters have resulted in decreased expression. This point is important because the expression plasmids should be made publicly available, and for their use to be understood properly, this must be clarified. The details of the plasmids are unclear. Perhaps listed in the table? - unclear. This aspect would be important for the field to effectively use the reagents.
5. This manuscript speculates several times that with more abundant PIs like PI45P2, the observed saturation effect is probably not happening. This should be removed. While the back of envelope calculations may reflect an ideal scenario, the heterogeneity of distribution and multiple key cellular structures involved would seem to corral increased PI45P2 levels in certain regions. These factors amid multivalency and electrostatic mechanisms of lipid effector recruitment (e.g.

MARCKS) suggest that speculation may be too strong. Moreover, Maib et al JCB 2024 demonstrated PI4P probe overexpression could directly mask the ability to detect PI4P post-fixation - not fully, but partially. Repeating the titration experiments of this manuscript for multiple PIs is entirely beyond the scope of reasonable, and hence, such experiments are not requested, in favor of adopting more conscientious speculation.

****Minor:****

1. Schematics throughout need simplification, enabling their enlargement.
2. Numerous spelling (Fig. 4 schemas) and capitalizations need fixing.
3. Pg 1 Famous is not appropriate wording
4. Fig. 1A statistical testing of microscopy quantifications absent (generally, throughout) and should be included.
5. Fig.1. In a transient transfection, the protein expression is not uniform. Please explain how you normalized the quantification.
6. Fig. 1D. EGFP expression levels increased with EGF stimulation. How is this possible?
7. Fig. 1D. The images have pS473 whereas the y-axis label on box plots has p473. Can these box plots be labelled separately for consistency?
8. Fig.1. T308 phosphorylation is mentioned in Figure 1, but only pS473 data is shown.
9. Fig.1 legend. 'Over-expression of PH-AKT is hypothesised to outcompete the endogenous AKT's PH domain'. Why do you need to state a hypothesis in the legend?
10. Fig.1E You stated that the PH-AKT R25C-EGFP is stimulated by EGF addition. However, the GFP signal looks the same in both unstimulated and stimulated. Could you please clarify? Are you sure that the stimulation worked?
11. You mention...that the AKT enzyme is activated by PDK1 and TORC2, which phosphorylate at residues T308 and S473, respectively. Phosphorylation is also known to occur on T450 at c-tail. Does this phosphorylation also contribute to its activation?
12. Fig. 1 scale bar in all images equivalent?
13. Pg. 1 paragraph 1 "we have argued..." vs. paragraph 3"...consider that an..." feels like arguing with themselves.
14. Pg. 1 para 3 what is RFC score - must explain
15. Discussion of numbers of PIP3 vs. effectors etc may not be appropriate for the introduction, as the points made by these calculations are already made in the previous paragraphs. May fit better in pg 6 Mitigating PIP3 titration... with an accompanying schematic.
16. Pg 2 "a neonGreen" not well defined, needs accurate description.
17. Fig 2C should give a unstimulated trajectory of puncta/100 μm^2 to compare with the stimulated
18. Fig 2C and F and G should be systematized for easier comparison. E.g. min vs seconds, 0 timepoint of EGF/rapa addition
19. Pg 5 "...and calibrated them..." unclear what is being calibrated, as the text later states that the histograms are fit to monomer/dimer/multimer model resulting in 98.1% in monomer. Minor point.
20. Explain why baselines in Fig2CFG are different
21. Fig. 2 has quantification with images; Fig. 3 has it separate. Make consistent.
22. Fig. 3B comes before images? Where are the images? Also, y-axis = Intensity (a.u.). Is intensity

just full image field? Or per cell? All very unclear.

23. Fig. 3C missing images

24. Fig 3 C needs brightness/contrast adjusted as images are nearly entirely black (zero values).

25. Fig 3C needs scale bar systemization

26. Fig 4 needs 4 panels A-D

27. Pg 6 5-OH phosphatases needs reference

28. Fig 5B, make images bigger

****Referees cross-commenting****

I have read the other reviews and find them entirely reasonable. My impression is we landed on similar general content that needs work, none of which is out of line. The importance and care taken in the author's work is uniformly lauded.

2. Significance:

Significance (Required)

This manuscript clearly and reasonably demonstrates that the commonly used PIP3 sensor can be titrated to low concentrations, at which it does not interfere with Akt translocation and activation. This work is a good technical reference for the field. Signal transduction and membrane biologists should be especially interested in the data. The reviewer/s have core expertise in phosphoinositides, protein biochemistry, cell biology, and membrane biophysics.

3. How much time do you estimate the authors will need to complete the suggested revisions:

Estimated time to Complete Revisions (Required)

(Decision Recommendation)

Cannot tell / Not applicable

4. Review Commons values the work of reviewers and encourages them to get credit for their work. Select 'Yes' below to register your reviewing activity at Web of Science Reviewer Recognition Service (formerly Publons); note that the content of your review will not be visible on Web of Science.

No

Review #2

1. Evidence, reproducibility and clarity:

Evidence, reproducibility and clarity (Required)

The authors characterize the inhibition of lipid second messenger mediated cell signaling through lipid biosensors that outcompete endogenous effector proteins. This is a very important study that as it quantitatively assesses an issue that many people suspected to exist, yet never properly characterized. This paper is therefore as much a service to the community as a research study in its own right and should be published without undue delay. I am glad that the authors decided to carry out this study & really appreciate their work.

I do however, have a number of suggestions that I think will make the manuscript stronger and can be readily implemented, mostly by reformulating and/or re-analysis of existing datasets. I've structured my comments by the datasets in the respective figures to follow the logic of the paper.

- Throughout the manuscript, statistical tests are missing, e.g. in figures 1C-F. This must be amended in the revised version. The authors are making a very quantitative point about buffering, data should be treated accordingly.

- I do not think that "PIP3 titration" is the best term to describe the observed effect. "Titration" usually implies the controlled modulation of a concentration, e. g. in analytical chemistry. I think either "competitive binding of PIP3" or "buffering of free PIP3" are more adequate.

Specific comments:Figure 1

- Why are data in 1D-Ff shown as median, with interquartile ranges and 10-90 percentile distance when everything else in the paper is mean +/- se? There might be a good reason for it, but I did not find it mentioned everywhere

- The authors should test, whether the difference between the +EGF conditions in 1D (EGFP) and 1F (PH-AktR25C-EGFP) is indeed statistically significant. If this observation holds up, what does it mean? Is the mutant still competing with endogenous Akt despite the much-reduced binding affinity? The authors should discuss.

- How were biosensor/GFP positive cells chosen? Did the authors choose a defined fluorescence intensity cut-off? I think that a pure manual selection is problematic from a methodological point of view as this may introduce biases. Since the authors use Fiji, they can also simply use the "Analyze particles" function, which allows to automatically segment cells from a thresholded image. By choosing the same threshold for all images, it would be ensured that all images are treated exactly the same way.

- I am missing a statement in the methods section that all images were acquired using the same settings.

- I recommend that the authors include a single cell correlation plot of EGFP fluorescence intensity vs AktpS473 intensity in Figure 1 D-F. This should be rather informative & make the concentration dependence clear.

- I further recommend that the authors look at alterations of baseline Akt activity in the presence of the biosensor. In the images it looks like there might be an effect, but this is then lost in the analysis due to the normalization.

- Please include zoomed image insets in Fig. 1D-F, in the current magnification one needs to zoom in quite a bit to see the effect in the raw data. It is a clear effect, but having a zoomed version would make for much easier reading.

- Up to the authors: I wonder whether it is possible to extract an IC50 value for the competitive inhibition of Akt by the respective biosensors. The transient expression gives the authors access to a wide range of expression levels at the single cell level, which could be quantified by counterstaining with a EGFP-nanobody at a different color (since the EGFP fluorophore went through the fixation process, it is likely unsuitable for quantification) and microscope calibration. Activity could be quantified as the ratio of observed and expected Akt-pS473 fluorescence (derived from the mean FI per cell from the EGFP control). This is not strictly necessary, but would be a beautiful quantitative experiment, give an easy-to-understand number & make the paper much stronger.

Specific comments:Figure 2

- Overall, compelling data. However, 25 molecules/100 μm^2 at maximal recruitment feels low. Assuming a total cell surface area of appr. 2000 μm^2 per cell and taking a baseline of 5 molecules/100 μm^2 into account, this would mean that only about 400 copies of Akt are recruited in response to a pretty robust stimulus. Is it possible that the association reaction of the split GFP is not complete under these conditions? I think that a direct measurement of intracellular endogenous Akt concentration is required to put these numbers into context.

Specific comments:Figure 3

- I think that the classification by plasmid dose does not make a lot of sense, as the resulting expression levels are rather similar. I suggest to pool all traces and calculate mean curves by actual expression levels using a binning approach (e.g. 0-50 au, 50-100 au and so on in raw intensity from Figure 3b). If there is an effect in the realized concentration regime, this should pick it up.

Specific comments:Figure 5

- These are very interesting data, in particular with regard to the underlying PIP3 dynamics. I agree with the conclusion of the authors that shielding of PIP3 from degradation is the likely culprit. What I would like to see here is actual kinetic fits - and different terms. On- and off-rate imply biosensor binding, but these are likely rather fast and not on the minute-timescale. The detected processes are much more likely to reflect production and degradation of PIP3 and that should be reflected in the terminology. For the fit: I think that a simple rate law for subsequent reactions ($[\text{PIP3}] = C(e^{-k_1 t} - e^{-k_2 t})$) will give good results and yield effective rate constants for PIP3 generation and degradation. This implies the quasi-steady state assumption for biosensor binding and implies that $[\text{PIP3}]$ is proportional to the biosensor bound $[\text{PIP3}]$, but these are reasonable assumptions to make.

André Nadler

2. Significance:

Significance (Required)

This is an important paper, analyses the effects of over-expressed lipid biosensors on cell signalling in some detail and will be of significant interest to a broad readership.

3. How much time do you estimate the authors will need to complete the suggested revisions:

Estimated time to Complete Revisions (Required)

(Decision Recommendation)

Between 1 and 3 months

4. Review Commons values the work of reviewers and encourages them to get credit for their work. Select 'Yes' below to register your reviewing activity at Web of Science Reviewer Recognition Service (formerly Publons); note that the content of your review will not be visible on Web of Science.

Yes

Review #3

1. Evidence, reproducibility and clarity:

Evidence, reproducibility and clarity (Required)

This is essentially a methods paper in which the authors provide a detailed and highly quantitative analysis of the potentially deleterious effects of expressing phosphoinositide-binding domains as biosensors. Specifically, they study the effects on PIP3 signalling, using biosensors that are widely used in the field.

They show that the most-commonly used method of expressing PIP3 biosensors using transient transfection with viral promoters has clear deleterious effects on downstream signalling due to out-competing the endogenous effectors. Importantly, they also describe a new approach to overcome this by developing new plasmids and methodology to express these reporters at low levels.

****Major comments:****

The work in this paper is thorough and very nicely done. I particularly appreciate the efforts to quantitate or estimate actual numbers and densities of molecules, which significantly strengthen their arguments. The data are excellent and strongly support all their conclusions. I would therefore be happy to see this work published in its current form.

****Minor comments:****

I only have some minor and optional suggestions for improvement.

In figure 1D-F they show that PH-Atk-EGFP expression can suppress downstream Akt activation by quantifying P-Akt signal by microscopy. In these panels they say they selectively measure this in GFP-expressing cells, but it is not clear how they define which cells are expressing GFP - was a

threshold used? Also, it would be nice to also measure both PH-Akt-GFP and P-Akt staining by flow cytometry to look for a correlation. Is there a threshold of biosensor expression that blocks downstream signalling, or is there a linear relationship? This might help specifically measure how much biosensor is too much.

Some of their microscopy images (e.g. Fig 1D-F, Fig 5) are very small and would benefit from a zoom box - especially when they are trying to demonstrate single molecule detection.

2. Significance:

Significance (Required)

This is both a methods paper and cautionary tale for cell biologists working in this field. Whilst everyone who uses these probes should be aware of the potential risk of biosensors titrating our effectors, this is often not sufficiently acknowledged. This paper is a very nice and clear demonstration of these risks, exemplified with probably the most highly-used biosensor and key downstream signalling pathway.

Whilst the concepts presented are not especially novel, this paper nonetheless makes an important contribution to the community and hopefully will make others more cautious in how they use these biosensors.

3. How much time do you estimate the authors will need to complete the suggested revisions:

Estimated time to Complete Revisions (Required)

(Decision Recommendation)

Less than 1 month

Yes

Manuscript number: RC-2024-02690

Corresponding author(s): Hammond, Gerald

1. *General Statements 1*
2. *Point-by-point description of the revisions 1*

Reviewer #1 (Evidence, reproducibility and clarity (Required)): 1

Reviewer 1 Major: 2

Reviewer 1 Referees cross-commenting** 8

Reviewer #1 (Significance (Required)): 8

Reviewer #2 (Evidence, reproducibility and clarity (Required)): 9

Reviewer 2 Major: 9

Reviewer #2 (Significance (Required)): 14

Reviewer #3 (Evidence, reproducibility and clarity (Required)): 14

Reviewer 3 Major comments: 14

Reviewer 3 Minor comments: 14

Reviewer #3 (Significance (Required)): 15

General Statements

We are extremely grateful to all three reviewers for their careful and considered comments on the manuscript. We have now addressed all of these points in a fully revised manuscript. We think that these have significantly improved the paper. We note that in addition to reviewer-requested changes and data, we have added additional data to the results presented in Figure 2E (which strengthens but does not change the original result).

1. Point-by-point description of the revisions

Reviewer #1 (Evidence, reproducibility and clarity (Required)):

The authors present the use of previously identified biosensors in a single-molecule concentration regime to address lipid effector recruitment. Using controlled and careful single-cell based analysis, the study investigates how expression of the commonly used PIP3 sensor based on Akt-PH domain interferes with the native detection of PIP3. Predominantly live-cell fluorescence microscopy coupled to image analysis drives their studies.

Conceptually, this manuscript carefully and quantitatively describes the influence of lipid biosensor overexpression and presents a means to overcome the inherent and long-recognized problems therein. This solution, namely employing low expression of the lipid biosensor, should be generally applicable. The work is of general interest to cell biologists focused on answering questions at

membranes and organelles, including especially those interested in lipid-mediated signaling transductions.

Reviewer 1 Major:

#1.1 The terminology "single molecule biosensor" is not really appropriate. A protein is not "single-molecule". An enzyme does not "single molecule". Better is biosensors at single-molecule expression levels. In most cases, this should be changed. Single-molecule vs single-cell vs. bulk measurements are often poorly defined in quantifications and conflating these does not help the case, which is already supported by generally clear data.

We appreciate the reviewer's thoughtful critique of our grammatically incorrect use of jargon; we saw this as soon as they mentioned it! We have amended the manuscript where appropriate as detailed:

- Title is now changed to "Lipid Biosensors Expressed at Single Molecule Levels Mitigates Inhibition of Endogenous Effector Proteins"
- Last paragraph of the **introduction** on **p. 2** now reads "As well as alleviating inhibition of PI3K signaling, biosensors expressed at these low levels show improved dynamic range and report more accurate kinetics than their over-expressed counterparts."
- The title of the results section on **p. 6** is now: **Mitigating PIP3 competition using biosensors expressed at single molecule levels**
- Last paragraph of the results section on **p.6** now reads: "this showed that when expressed at single molecule levels, the biosensor has substantially better dynamic range".

#1.2 Figure 1D-F, images not as clearly describing quantitation as one would hope. Untransfected cells in 1E should demonstrate more translocated Akt-pS473 than transfected, but it is difficult for this reviewer to find. Consider inset images in addition to the wider field. Consider also moving the "negative" data of Fig 1B-C to Supplement.

We regret not making this figure easier to interpret; we have substantially updated the figure, as comprehensively detailed in our point-by-point response to reviewer 2's point 2.3. To specifically address this reviewer's concerns:

The older figure used non-confocal, low-resolution images that were used for quantification. Such an approach was employed to enable fluorescence from the entire cellular volume to

be captured, which produces more robust quantification. However, to the reviewer's point, it is not possible to see the translocation of PH-AKT1 nor translocated AKT-pS473 in these images. Fortunately, we had in parallel captured high resolution confocal images for some experiments. These are now shown in **Fig 1D-E**, which clearly shows translocated AKT-pS473 and PH-AKT-EGFP

#1.3 *The cell line being used is not clearly specified after the initial development of the NG1 followed by CRISPRed NG2 onto Akt. For example, for the Figure 3C experiments, the text states "complete ablation of endogenous AKT1-NG2" but this information is not apparent from the figure legend or figure. Throughout the cell line used and the aspects transfected need to be made explicitly clear.*

We are grateful to the reviewer for highlighting this ambiguity. We have now defined the gene-edited cells used throughout as "AKT1-NG2 cells" and expressly used this term when referring to experiments in figures 2-5.

#1.4 *Fig. 5 shows single cells. It is therefore unclear if broken promoters have resulted in decreased expression. This point is important because the expression plasmids should be made publicly available, and for their use to be understood properly, this must be clarified. The details of the plasmids are unclear. Perhaps listed in the table? - unclear. This aspect would be important for the field to effectively use the reagents.*

Thank you for drawing our attention to the lack of adequate detail here. We have now updated the results text to expressly reference Morita et al., 2022 where the origins of the truncated CMV promoters are detailed. We have also updated the plasmids table 1 to add pertinent details for these constructs: *pCMVd3 plasmids are based on the pEGFP-C1 backbone, with the CMV promoter truncated to remove 18 of the 26 putative transcription factor binding sites in the human Cytomegalovirus Major Intermediate Enhancer/Promoter (pCMV Δ 3 as described in Morita et al., 2012). The full sequences will be deposited with the plasmids on Addgene.

We did not perform a formal comparison of full vs truncated promoters. Our only observation is that the truncated promoters greatly help in increasing the number of expressing cells presenting single-molecule resolvable expression levels (though the approach can still work with full promoters).

#1.5 *This manuscript speculates several times that with more abundant PIs like PI45P2, the observed saturation effect is probably not happening. This should be removed. While the back of envelope calculations may reflect an ideal scenario, the heterogeneity of distribution and multiple key cellular structures involved would seem to corral increased PI45P2 levels in certain regions. These factors amid multivalency and electrostatic mechanisms of lipid effector recruitment (e.g. MARCKS) suggest that speculation may be too strong. Moreover, Maib et al JCB 2024 demonstrated PI4P probe overexpression could directly mask the ability to detect PI4P post-fixation - not fully, but partially. Repeating the titration experiments of this manuscript for multiple PIs is entirely beyond the scope of reasonable, and hence, such experiments are not requested, in favor of adopting more conscientious speculation.*

The reviewer's point is well taken. Whilst we still believe the overall argument for lipids is sounds (for example, PS or cholesterol are far too abundant for any expressed, stoichiometric binding protein to bind the majority of the population) even abundant phosphoinositides like PI4P and PI(4,5)P₂ are an edge case. We have therefore undated the **first paragraph** of the **introduction** on **p. 1** to be less explicit: *One of the most prominent is the fact that lipid engagement by a biosensor occludes the lipid's headgroup, blocking its interaction with proteins that mediate biological function. It follows that large fractions of lipid may be effectively outcompeted by the biosensor, inhibiting the associated physiology. We have argued that, in most cases, this is unlikely because the total number of lipid molecules outnumbers expressed biosensors by one to two orders of magnitude (Wills et al., 2018). However, for less abundant lipids, total molecule copy numbers may be in the order of tens to hundreds of thousands, making competition by biosensors a real possibility.*

We also removed the explicit discussion of PI(4,5)P₂ from the introduction, and focus now solely on the PI3K lipids.

Reviewer 1 Minor:

#1.6 *Schematics throughout need simplification, enabling their enlargement.*

We have now enlarged the size of all schematics

#1.7 *Numerous spelling (Fig. 4 schemas) and capitalizations need fixing.*

Thank you for drawing our attention to these. We have thoroughly proof-read the figure panels and corrected errors.

#1.8 *Pg 1 Famous is not appropriate wording*

We respectfully beg to differ with the reviewer here. We believe it is perfectly accurate to state that PIP₃ is a second messenger molecule that is known about by many people; we see this as the dictionary definition of the word "famous".

#1.9 *Fig. 1A statistical testing of microscopy quantifications absent (generally, throughout) and should be included.*

This was indeed an oversight on our part. We have now added appropriate multiple comparisons tests to the data presented in figures 1F, 3F, 4C, 4F and 5C.

#1.10 *Fig. 1. In a transient transfection, the protein expression is not uniform. Please explain how you normalized the quantification.*

We hope this is now clarified by the expanded "**Image Analysis**" part of the methods section on **pp. 10-11** (relevant sentence is underlined): *For immunofluorescence, we identified individual cells by auto thresholding the DAPI channel using the "Huang" method, followed by the Watershed function to segment bunched cells that appeared to touch. We then used the Voronoi function to generate boundary lines for the segmentation of the cells. To identify cytoplasm, auto thresholding of the CellMask channel using the "Huang"*

function was employed, with the cells segmented by adding the nuclear Voronoi boundaries. The “analyze particles” function was then used to identify individual cellular ROIs that were greater than $10 \mu\text{m}^2$ and were not touching the image periphery. These ROIs were used to measure the raw 12-bit intensity of the EGFP and AKT-pS473 channels. A cutoff of EGFP > 100 was used to define EGFP-positive cells, since this value was greater than the mean ± 3 standard deviations of the non-transfected cells’ EGFP intensity. Background intensity of AKT-pS473 was estimated from control cells subject to immunofluorescence in the absence of AKT-pS473 antibody; this value was subtracted from the measured values of all other conditions.

#1.11 Fig. 1D. EGFP expression levels increased with EGF stimulation. How is this possible?

There appeared to be a difference due to the presence of 5 strongly expressing cells in the chosen field in the original field for the EGF stimulated, EGFP cells. However, this arose just by chance. The new set of high-resolution images in the new figure 1 were selected to be more representative.

#1.12 Fig. 1D. The images have pS473 whereas the y-axis label on box plots has p473. Can these box plots be labelled separately for consistency?

Thank you. This has now been corrected in the revised Figure 1.

#1.13 Fig. 1. T308 phosphorylation is mentioned in Figure 1, but only pS473 data is shown.

Both T308 and S473 phosphorylation are indicative of AKT activation. However, antibodies suitable for immunofluorescence are only available for pS473, hence why our experiments are restricted to this moiety.

#1.14 Fig. 1 legend. 'Over-expression of PH-AKT is hypothesised to outcompete the endogenous AKT's PH domain'. Why do you need to state a hypothesis in the legend?

We included this statement for the benefit of the casual reader – i.e. one who looks at the pictures, but doesn't read the main text!

#1.15 Fig. 1E You stated that the PH-AKT R25C-EGFP is stimulated by EGF addition. However, the GFP signal looks the same in both unstimulated and stimulated. Could you please clarify? Are you sure that the stimulation worked?

We have clarified the **second paragraph** of the results section “**Inhibition of AKT activation by PIP3 biosensor**” on p. 4 as follows: *In the non PIP₃ binding PH-AKT1^{R25C}-EGFP positive cells, we still observed an increase in pS473 intensity.*

The revised figure 1 images also show that PH-AKT1^{R25C} does not translocate to the membrane with EGF stimulation.

#1.16 You mention...that the AKT enzyme is activated by PDK1 and TORC2, which phosphorylate at residues T308 and S473, respectively. Phosphorylation is also known to occur on T450 at c-tail. Does this phosphorylation also contribute to its activation?

Yes and no. Threonine 450 phosphorylation is thought to occur co-translationally and is important for AKT stability (see Truebestein et al as cited in the manuscript). It is not really relevant in the context for T308 and S473, which are phosphorylated acutely to activate the protein.

#1.17 *Fig. 1 scale bar in all images equivalent?*

We have now added scale bars to panels in both figure 1D and E to clarify.

#1.18 *Pg. 1 paragraph 1 "we have argued..." vs. paragraph 3"...consider that an..." feels like arguing with themselves.*

We believe the re-write we have done in response to major point **#1.5** clarifies this point also.

#1.19 *Pg. 1 para 3 what is RFC score - must explain*

We have now defined this more clearly in **third** paragraph of the **introduction** on **p. 1**: **PH domain containing PIP₃ effector proteins can be predicted based on sequence comparison to known PIP₃ effectors vs non effectors using a recursive functional classification matrix for each amino acid (Park et al., 2008).**

#1.20 *Discussion of numbers of PIP3 vs. effectors etc may not be appropriate for the introduction, as the points made by these calculations are already made in the previous paragraphs. May fit better in pg 6 Mitigating PIP3 titration... with an accompanying schematic.*

Respectfully, we prefer to keep this discussion of molecular concentrations, as this adds details and specifics to the pathway that is core to the paper.

#1.21 *Pg 2 "a neonGreen" not well defined, needs accurate description.*

We have clarified this in the sentence in the **first paragraph** of the **results** section "**Genomic tagging of AKT1...**" on **p. 4**, which includes the citation to the full description of the tag: **To that end, we used gene editing to incorporate a bright, photostable neonGreen fluorescent protein to the C-terminus of AKT1 via gene editing using a split fluorescent protein approach (Kamiyama et al., 2016).**

#1.22 *Fig 2C should give a unstimulated trajectory of puncta/100 um2 to compare with the stimulated*

Unfortunately, we did not record a full 5.5-minute video-rate time-lapse with unstimulated cells. However, we do not believe this control is essential for this experiment, since this example data is included to illustrate (1) the problem of photobleaching, which is clear in the 30-s pre-stimulus and (2) the variability in the raw molecule counts.

#1.23 Fig 2C and F and G should be systematized for easier comparison. E.g. min vs seconds, 0 timepoint of EGF/rapa addition

We have made the adjustment to figure 2C to be consistent with 2F and G:

#1.24 Pg 5 "...and calibrated them..." unclear what is being calibrated, as the text later states that the histograms are fit to monomer/dimer/multimer model resulting in 98.1% in monomer. Minor point.

We have clarified this point in the **second paragraph** of the **results** section "**Genomic tagging of AKT1...**" on **p. 4** as follows: We analyzed the intensity of these spots and compared them to intensity distributions from a known monomeric protein localized to the plasma membrane (PM) and expressed at single molecule levels

#1.25 Explain why baselines in Fig2CFG are different

We did not comment on figure 2C; it is a single cell measurement, as opposed to the mean of 20 cells reported in F. However, we do now clarify the difference between figure 2F and G as the very end of the "**Genomic tagging of AKT1...**" **results** section on **p 4**: Notably, baseline AKT-NG2 localization increased from ~5 to ~15 per 100 μm^2 in iSH2 cells, perhaps because the iSH2 construct does not contain the inhibitory SH2 domains of p85 regulatory subunits, producing higher basal PI3K activity.

#1.26 Fig. 2 has quantification with images; Fig. 3 has it separate. Make consistent.

We sometimes combine images with quantification, and other times separate the panel containing graphs. This is done deliberately, depending on whether the reader is directed to both together, or whether we consider the data separately in the results section.

#1.27 Fig. 3B comes before images? Where are the images? Also, y-axis = Intensity (a.u.). Is intensity just full image field? Or per cell? All very unclear.

We have modified both the graph y-axis label and the figure legend to clarify: (C) TIRF imaging of AKT1-NG2 cells from (B) stimulated with 10 ng/ml EGF

#1.28 Fig. 3C missing images

We believe the reviewer is referring to the mCherry channel for the "0 ng cDNA" condition. These images are missing because they do not exist. Since these cells were transfected with pUC19, there was no mCherry fluorescence to image.

#1.29 *Fig 3 C needs brightness/contrast adjusted as images are nearly entirely black (zero values).*

We believe the addition of insets addresses this concern. To the reviewer's specific suggestion, we found that further increases in the brightness and contrast will bring up the camera noise, but this then occludes the signal from single molecules, such as those found after EGF stimulation of the 0 ng condition.

#1.30 *Fig 3C needs scale bar systemization*

We believe that the incorporation of scaled 6 μm insets addresses this point.

#1.31 *Fig 4 needs 4 panels A-D*

We have now added these individual panel labels to **figure 4**.

#1.32 *Pg 6 5-OH phosphatases needs reference*

We have added a citation to Trésaugues at the very end of the “**Sequestration of PIP₃ by lipid biosensors**” **results** section on **p. 6**, which describes the activity of the whole 5-OH phosphatase activity against PIP₃, not just the SHIP phosphatases.

#1.33 *Fig 5B, make images bigger*

Again, we trust that the addition of insets to all single molecule images has addressed this point.

*Reviewer 1 Referees cross-commenting***

I have read the other reviews and find them entirely reasonable. My impression is we landed on similar general content that needs work, none of which is out of line. The importance and care taken in the author's work is uniformly lauded.

We agree. At the risk of restoring to alliteration, we have been delighted to receive a trio of clear, concise and consistent comments on the manuscript! We believe it is now much improved.

Reviewer #1 (Significance (Required)):

This manuscript clearly and reasonably demonstrates that the commonly used PIP3 sensor can be titrated to low concentrations, at which it does not interfere with Akt translocation and activation. This work is a good technical reference for the field. Signal transduction and membrane biologists should be especially interested in the data. The reviewer/s have core expertise in phosphoinositides, protein biochemistry, cell biology, and membrane biophysics.

Reviewer #2 (Evidence, reproducibility and clarity (Required)):

The authors characterize the inhibition of lipid second messenger mediated cell signaling through lipid biosensors that outcompete endogenous effector proteins. This is a very important study that as it quantitatively assesses an issue that many people suspected to exist, yet never properly characterized. This paper is therefore as much a service to the community as a research study in its own right and should be published without undue delay. I am glad that the authors decided to carry out this study & really appreciate their work.

I do however, have a number of suggestions that I think will make the manuscript stronger and can be readily implemented, mostly by reformulating and/or re-analysis of existing datasets. I've structured my comments by the datasets in the respective figures to follow the logic of the paper.

Reviewer 2 Major:

#2.1 *Throughout the manuscript, statistical tests are missing, e.g. in figures 1C-F. This must be amended in the revised version. The authors are making a very quantitative point about buffering, data should be treated accordingly.*

We have now added appropriate multiple comparisons tests to figures 1F, 3F, 4C, 4F and 5C.

#2.2 *I do not think that "PIP3 titration" is the best term to describe the observed effect. "Titration" usually implies the controlled modulation of a concentration, e. g. in analytical chemistry. I think either "competitive binding of PIP3" or "buffering of free PIP3" are more adequate.*

This point is well taken. We have now replaced the word "titration" throughout, replacing it with either "competitive binding" or "sequestration".

#2.3 *Specific comments: Figure 1*

#2.3a *Why are data in 1D-Ff shown as median, with interquartile ranges and 10-90 percentile distance when everything else in the paper is mean +/- se? There might be a good reason for it, but I did not find it mentioned everywhere*

For consistency's sake, we have changed figure 1F to show a bar graph, though as noted in the figure legend: Graphs show medians \$\pm\$ 95% confidence interval of the median from 82-160 cells pooled from three experiments (medians are reported since the data are not normally distributed).

#2.3b *The authors should test, whether the difference between the +EGF conditions in 1D (EGFP) and 1F (PH-AktR25C-EGFP) is indeed statistically significant. If this observation holds up, what does it mean? Is the mutant still competing with endogenous Akt despite the much-reduced binding affinity? The authors should discuss.*

We have re-analyzed the data in figure 1, with the quantitative data presented in figure 1F combined with statistical analysis. The new data shows no significant effect of the PH-AKT1^{R25C} mutant in either resting or EGF stimulated condition

These results are also described in the **second paragraph** of the first **results** section on **pp. 3-4**: *This analysis showed that the R25C mutant had no substantial effect on pS473 levels, whereas wild-type PH-AKT greatly inhibited pS473 staining in EGF-stimulated cells as well as reducing basal levels in serum starved cells (Fig. 1F).*

#2.3c *How were biosensor/GFP positive cells chosen? Did the authors choose a defined fluorescence intensity cut-off? I think that a pure manual selection is problematic from a methodological point of view as this may introduce biases. Since the authors use Fiji, they can also simply use the "Analyze particles" function, which allows to automatically segment cells from a thresholded image. By choosing the same threshold for all images, it would be ensured that all images are treated exactly the same way.*

We had initially opted for manual outlining of cells since automatic segmentation of irregularly-shaped HEK293a cells is imperfect. However, we agree with André that this opens the possibility of bias. We have therefore re-run the analysis with an automated segmentation and thresholding approach, as suggested. This is detailed in the **second paragraph** of the first **results** section on **pp. 3-4**: *In parallel, we imaged cells with a low resolution 0.75 NA air objective to capture fluorescence from the cells' entire volume, then quantified these images using an automatically determined threshold for GFP-positive cells (see Materials and Methods). This analysis showed that the R25C mutant had no substantial effect on pS473 levels, whereas wild-type PH-AKT greatly inhibited pS473 staining in EGF-stimulated cells as well as reducing basal levels in serum starved cells (Fig. 1F).*

Further detail is provided in the **first paragraph** of the **"Image analysis"** subsection of the methods on **pp. 10-11**: *For immunofluorescence, we identified individual cells by auto thresholding the DAPI channel using the "Huang" method, followed by the Watershed function to segment bunched cells that appeared to touch. We then used the Voronoi function to generate boundary lines for the segmentation of the cells' cytoplasm. To identify cytoplasm, auto thresholding of the CellMask channel using the "Huang" function was employed, with the images segmented by adding the nuclear Voronoi boundaries. The "analyze particles" function was then used to identify individual cellular ROIs that were greater than 10 μm² and were not touching the image periphery. These ROIs were used to measure the raw 12-bit intensity of the EGFP and AKT-pS473 channels. A cutoff of EGFP >*

100 was used to define EGFP-positive cells, since this value was greater than the mean \pm 3 standard deviations of the untransfected cells' EGFP intensity. Background intensity of AKT-pS473 was estimated from control cells subject to immunofluorescence with the AKT-pS473 antibody omitted; this value was subtracted from the measured values of all other conditions.

#2.3d *I am missing a statement in the methods section that all images were acquired using the same settings.*

This was indeed an important oversight on our part – thanks for spotting the omission of this crucial detail. This is now included at the end of the “**Immunofluorescence**” section of the Methods on **pp. 9-10**: **Identical laser excitation power, scan speeds and photomultiplier gains were used across experiments to enable direct comparison.**

#2.3e *I recommend that the authors include a single cell correlation plot of EGFP fluorescence intensity vs Akt-pS473 intensity in Figure 1 D-F. This should be rather informative & make the concentration dependence clear.*

We did not observe a strong correlation between PH-AKT1-EGFP intensity and pS473 staining, likely driven by both the imprecision of the cell segmentation and the fact that very low concentrations of PH domain effectively inhibit endogenous AKT1 (as we show in the later figures with the more precise, live cell AKT-NG2 recruitment experiments: see response to **#2.5**).

#2.3f *I further recommend that the authors look at alterations of baseline Akt activity in the presence of the biosensor. In the images it looks like there might be an effect, but this is then lost in the analysis due to the normalization.*

As covered in our response to **#2.3b**, there is indeed an inhibition of baseline pS473 in PH-AKT1-EGFP expressing cells, now explicitly quantified and documented in results.

#2.3g *Please include zoomed image insets in Fig. 1D-F, in the current magnification one needs to zoom in quite a bit to see the effect in the raw data. It is a clear effect, but having a zoomed version would make for much easier reading.*

We now include high-resolution confocal images instead of low power, low NA volumes as shown in the last version of the manuscript, which we believe addresses this point and also **reviewer #1.2**.

#2.3h *Up to the authors: I wonder whether it is possible to extract an IC50 value for the competitive inhibition of Akt by the respective biosensors. The transient expression gives the authors access to a wide range of expression levels at the single cell level, which could be quantified by counterstaining with a EGFP-nanobody at a different color (since the EGFP fluorophore went through the fixation process, it is likely unsuitable for quantification) and microscope calibration. Activity could be quantified as the ratio of observed and expected Akt-pS473 fluorescence (derived from the mean FI per cell from the EGFP control). This is not strictly necessary, but would be a beautiful quantitative experiment, give an easy-to-understand number & make the paper much stronger.*

This is a great suggestion, but does not produce precise enough data to work out, as we detail in response to **#2.3e**. From our data in new figure 3F and figure 5, it seems we have not explored the appropriate expression range to see intermediate levels of inhibition necessary to estimate IC₅₀. This would be a cool experiment though!

#2.4 *Specific comments: Figure 2. Overall, compelling data. However, 25 molecules/100 μm² at maximal recruitment feels low. Assuming a total cell surface area of appr. 2000 μm² per cell and taking a baseline of 5 molecules/100 μm² into account, this would mean that only about 400 copies of Akt are recruited in response to a pretty robust stimulus. Is it possible that the association reaction of the split GFP is not complete under these conditions? I think that a direct measurement of intracellular endogenous Akt concentration is required to put these numbers into context.*

This is an excellent point that we had missed. We now specifically address this point in the **third paragraph of the “Genomic tagging of AKT...” section on p. 4: Accumulation of AKT-NG2 was ~25 molecules per 100 μm², which assuming a surface area of ~1,500 μm² per cell corresponds to ~375 molecules total. It should be noted that tagging likely only occurred at a single allele in each cell, and the population still exhibited expression of non-edited AKT1 (Fig. 2B). Given that HEK293 are known to be pseudotriploid (Bylund et al., 2004), the true number of AKT1 molecules would be at least 1,125. However, given an estimated total copy number of 23,000 AKT1 in these cells (Cho et al., 2022), this is still only about 5%. However, we do not interpret these raw numbers due to uncertainties in the efficiency of NG2 complementation under these conditions, as well as potential for reduced expression from the edited allele.**

We also removed the specific comment on molecule density from the abstract.

#2.5 *Specific comments: Figure 3 I think that the classification by plasmid dose does not make a lot of sense, as the resulting expression levels are rather similar. I suggest to pool all traces and calculate mean curves by actual expression levels using a binning approach (e.g. 0-50 au, 50-100 au and so on in raw intensity from Figure 3b). If there is an effect in the realized concentration regime, this should pick it up.*

This is an excellent suggestion, and we have done just that: thank you! The data is now included as a new panel **Fig. 3F**. The result is described in the results section, “Sequestration of PIP₃ by lipid biosensors”, end of the **first paragraph on pp. 4-6: To observe the concentration-dependence of AKT1-PH-mCherry inhibition, we pooled the single cell data from these experiments and split transfected cells into cohorts based on raw expression level (excitation and gain were consistent between experiments, allowing direct comparison). This analysis showed profound inhibition of AKT1-NG2**

recruitment at all expression levels, with a slightly reduced effect only visible in the lowest expressing cohort (**Fig. 2F**).

#2.6 Specific comments: *Figure 5 These are very interesting data, in particular with regard to the underlying PIP3 dynamics. I agree with the conclusion of the authors that shielding of PIP3 from degradation is the likely culprit. What I would like to see here is actual kinetic fits - and different terms. On- and off-rate imply biosensor binding, but these are likely rather fast and not on the minute-timescale. The detected processes are much more likely to reflect production and degradation of PIP3 and that should be reflected in the terminology. For the fit: I think that a simple rate law for subsequent reactions ($[PIP3]=C(e^{-k_1t}-e^{-k_2t})$) will give good results and yield effective rate constants for PIP3 generation and degradation. This implies the quasi-steady state assumption for biosensor binding and implies that $[PIP3]$ is proportional to the biosensor bound $[PIP3]$, but these are reasonable assumptions to make.*

The is an excellent suggestion, which we have added. Specifically, fits are now present on **Figs. 5G** and **5I**; we describe these in the **last paragraph of results on p. 8**: **Normalizing data from both expression modes to their maximum response (Fig. 5G)** and fitting kinetic profiles for cooperative synthesis and degradation reactions revealed the rate of synthesis is remarkably similar: 1.09 min^{-1} (95% C.I. 1.02-1.17) for single molecule expression vs 1.02 min^{-1} (95% C.I. 0.98-1.06) for over-expression. On the other hand, degradation slowed with over expression from 0.34 min^{-1} (95% C.I. 0.24-0.58) to 0.13 min^{-1} (95% C.I. 0.12-0.15). This is expected, since synthesis of PIP₃ molecules would not be prevented by biosensor. On the other hand, PIP₃ degradation could be slowed by the over-expressed biosensor competing with PTEN and 5-OH phosphatases that degrade PIP₃. An even more exaggerated result is achieved with the cPHx1 PI(3,4)P₂ biosensor; this shows an increase in fold-change over baseline of 600% for single molecule expression levels, compared to only 100% in over-expressed cells (**Fig. 5H**). Again, the degradation rate of the signal is substantially slowed by the over-expressed sensor, reducing from 0.27 min^{-1} (95% C.I. 0.22-0.39) to 0.16 min^{-1} (95% C.I. 0.14-0.19), whereas synthesis remains only minorly impacted, changing from 0.61 min^{-1} (95% C.I. 0.57-0.64) to 0.54 min^{-1} (95% C.I. 0.52-0.56) with over-expression (**Fig. 5I**). Collectively, these data show that single molecule based PI3K biosensors show improved dynamic range and kinetic fidelity compared to the same sensors over-expressed.

Details of the fits are given in a new **methods section on p. 11**:

Fitting of reaction kinetics

Curve fitting was performed in Graphpad Prism 9 or later. For the data presented in Figs. 5G and 5I, both synthesis and degradation phases displayed clear “s” shaped profiles not well fit by simple first order kinetics. Since activation of the PI3K pathway involves many multiplicative interactions between adapters and allosteric activation of the enzymes themselves, we assumed cooperativity and fit reactions with the two phase reaction as follows:

$$F_t = \begin{cases} F_0, & x < 0 \\ F_{max} \left(\frac{t^{n_{syn}}}{t^{n_{syn}} + \tau_{syn}^{n_{syn}}} - \frac{t^{n_{deg}}}{t^{n_{deg}} + \tau_{deg}^{n_{deg}}} \right), & x \geq 0 \end{cases}$$

Where F_t denotes $\Delta F_t / \Delta F_{MAX}$, n_{syn} and n_{deg} are the Hill coefficients of the respective synthesis and degradation reactions, and the rate constants for the reactions are derived from $k_{syn} = 1/\tau_{syn}$ and $k_{deg} = 1/\tau_{deg}$.

André Nadler

Reviewer #2 (Significance (Required)):

This is an important paper, analyses the effects of over-expressed lipid biosensors on cell signalling in some detail and will be of significant interest to a broad readership.

Reviewer #3 (Evidence, reproducibility and clarity (Required)):

This is essentially a methods paper in which the authors provide a detailed and highly quantitative analysis of the potentially deleterious effects of expressing phosphoinositide-binding domains as biosensors. Specifically, they study the effects on PIP3 signalling, using biosensors that are widely used in the field.

They show that the most-commonly used method of expressing PIP3 biosensors using transient transfection with viral promoters has clear deleterious effects on downstream signalling due to out-competing the endogenous effectors. Importantly, they also describe a new approach to overcome this by developing new plasmids and methodology to express these reporters at low levels.

Reviewer 3 Major comments:

The work in this paper is thorough and very nicely done. I particularly appreciate the efforts to quantitate or estimate actual numbers and densities of molecules, which significantly strengthen their arguments. The data are excellent and strongly support all their conclusions. I would therefore be happy to see this work published in its current form.

Reviewer 3 Minor comments:

I only have some minor and optional suggestions for improvement.

#3.1 *In figure 1D-F they show that PH-Akt-EGFP expression can suppress downstream Akt activation by quantifying P-Akt signal by microscopy. In these panels they say they selectively measure this in GFP-expressing cells, but it is not clear how they define which cells are expressing GFP - was a threshold used? Also, it would be nice to also measure both PH-Akt-GFP and P-Akt staining by flow cytometry to look for a correlation. Is there a threshold of biosensor expression that blocks downstream signalling, or is there a linear relationship? This might help specifically measure how much biosensor is too much.*

This is an important comment, also raised by reviewer 2. We provide a detailed explanation and outline revisions that address this in our response to reviewer **#2.3c**; essentially, we replaced the analysis with an automated segmentation and quantification, estimating GFP-

positive cells from a fraction of non transfected cells. We have not performed a FACS analysis, but as we note in our response to **#2.3e** and **#2.3h**, the correlation between EGFP and pAKT staining is imprecise in these experiments. The new **Fig. 3C** does address this point for AKT1-NG2 recruitment, as described in our response to **#2.5**.

#3.2 *Some of their microscopy images (e.g. Fig 1D-F, Fig 5) are very small and would benefit from a zoom box - especially when they are trying to demonstrate single molecule detection.*

This is a fair point raised by all of the reviewers in one form or another. We have added zoomed insets to all of the single molecule images in Figs 2-5, and added higher magnification, confocal section images to Fig. 1.

Reviewer #3 (Significance (Required)):

This is both a methods paper and cautionary tale for cell biologists working in this field. Whilst everyone who uses these probes should be aware of the potential risk of biosensors titrating our effectors, this is often not sufficiently acknowledged. This paper is a very nice and clear demonstration of these risks, exemplified with probably the most highly-used biosensor and key downstream signalling pathway.

Whilst the concepts presented are not especially novel, this paper nonetheless makes an important contribution to the community and hopefully will make others more cautious in how they use these biosensors.

January 7, 2025

RE: JCB Manuscript #202412026T

Gerald Hammond
University of Pittsburgh School of Medicine

Dear Dr. Hammond:

Thank you for submitting your revised manuscript entitled "Lipid Biosensors Expressed at Single Molecule Levels Mitigate Inhibition of Endogenous Effector Proteins". Your paper has been assessed by the original reviewers from Review Commons who all appreciate your thorough revisions and responses to their comments. Editorially, we find that your study addresses an important outstanding issue for the community, therefore we would be happy to publish your paper in JCB pending final revisions necessary to meet our formatting guidelines (see details below).

A. MANUSCRIPT ORGANIZATION AND FORMATTING:

- 1) Text limits: Character count for Articles is < 40,000, not including spaces. Count includes abstract, introduction, results, discussion, and acknowledgments. Count does not include title page, figure legends, materials and methods, references, tables, or supplemental legends.
- 2) Figures limits: Articles may have up to 10 main text figures.
- 3) Figure formatting: Scale bars must be present on all microscopy images, including inset magnifications. Molecular weight or nucleic acid size markers must be included on all gel electrophoresis. Aspect ratios of images may not be altered.
- 4) Statistical analysis: Error bars on graphic representations of numerical data must be clearly described in the figure legend. The number of independent data points (n) represented in a graph must be indicated in the legend. Statistical methods should be explained in full in the materials and methods. For figures presenting pooled data the statistical measure should be defined in the figure legends. Please also be sure to indicate the statistical tests used in each of your experiments (either in the figure legend itself or in a separate methods section) as well as the parameters of the test (for example, if you ran a t-test, please indicate if it was one- or two-sided, etc.). Also, if you used parametric tests, please indicate if the data distribution was tested for normality (and if so, how). If not, you must state something to the effect that "Data distribution was assumed to be normal but this was not formally tested."
- 5) Abstract and title: The abstract should be no longer than 160 words and should communicate the significance of the paper for a general audience. The title should be less than 100 characters including spaces. Make the title concise but accessible to a general readership.
- 6) Materials and methods: Should be comprehensive and not simply reference a previous publication for details on how an experiment was performed. Please provide full descriptions in the text for readers who may not have access to referenced manuscripts.
- 7) All antibodies, cell lines, animals, and tools used in the manuscript should be described in full, including accession numbers for materials available in a public repository such as the Resource Identification Portal. Please be sure to provide the sequences for all of your primers/oligos and RNAi constructs in the materials and methods. You must also indicate in the methods the source, species, and catalog numbers (where appropriate) for all of your antibodies. Please also indicate the acquisition and quantification methods for immunoblotting/western blots.
- 8) Microscope image acquisition: The following information must be provided about the acquisition and processing of images:
 - a. Make and model of microscope
 - b. Type, magnification, and numerical aperture of the objective lenses
 - c. Temperature
 - d. Imaging medium
 - e. Fluorochromes
 - f. Camera make and model
 - g. Acquisition software

h. Any software used for image processing subsequent to data acquisition. Please include details and types of operations involved (e.g., type of deconvolution, 3D reconstitutions, surface or volume rendering, gamma adjustments, etc.).

10) Supplemental materials: There are strict limits on the allowable amount of supplemental data. Articles may have up to 5 supplemental figures. Please also note that tables, like figures, should be provided as individual, editable files. A summary of all supplemental material should appear at the end of the Materials and methods section.

13) ORCID IDs: ORCID IDs are unique identifiers allowing researchers to create a record of their various scholarly contributions in a single place. Please note that ORCID IDs are now *required* for all authors. At resubmission of your final files, please be sure to provide your ORCID ID and those of all co-authors.

Please note that JCB now requires authors to submit Source Data used to generate figures containing gels and Western blots with all revised manuscripts. This Source Data consists of fully uncropped and unprocessed images for each gel/blot displayed in the main and supplemental figures. Since your paper includes cropped gel and/or blot images, please be sure to provide one Source Data file for each figure that contains gels and/or blots along with your revised manuscript files. File names for Source Data figures should be alphanumeric without any spaces or special characters (i.e., SourceDataF#, where F# refers to the associated main figure number or SourceDataFS# for those associated with Supplementary figures). The lanes of the gels/blots should be labeled as they are in the associated figure, the place where cropping was applied should be marked (with a box), and molecular weight/size standards should be labeled wherever possible.

Journal of Cell Biology now requires a data availability statement for all research article submissions. These statements will be published in the article directly above the Acknowledgments. The statement should address all data underlying the research presented in the manuscript. Please visit the JCB instructions for authors for guidelines and examples of statements at (<https://rupress.org/jcb/pages/editorial-policies#data-availability-statement>).

B. FINAL FILES:

****The license to publish form must be signed before your manuscript can be sent to production. A link to the electronic license to publish form will be sent to the corresponding author only. Please take a moment to check your funder requirements before choosing the appropriate license.****

Thank you for your attention to these final processing requirements. Please revise and format the manuscript and upload materials within 7 days. If you need an extension for whatever reason, please let us know and we can work with you to determine a suitable revision period.

Thank you for this interesting contribution, we look forward to publishing your paper in Journal of Cell Biology.

Sincerely,

Lois Weisman, PhD
Monitoring Editor

Andrea L. Marat, PhD
Deputy Editor

Journal of Cell Biology

Reviewer #1 (Comments to the Authors (Required)):

The authors made satisfactory changes in response to the Reviewer Commons comments. They took these very seriously. The science is clearly presented, described, and claims made are well evidenced.

Reviewer #2 (Comments to the Authors (Required)):

I stand with my initial, very positive assessment during the Review Commons process, please see my original review report for specifics. In the meantime, the authors have fully and sufficiently addressed my comments during the revision and the resulting final version of the paper is a very strong manuscript that should be accepted without the need of further changes and in my opinion should be highlighted by a specialist in the field, since it comprehensively addresses the very important point of signalling lipid buffering by widely used biosensors in a quantitative fashion.
André Nadler

Reviewer #3 (Comments to the Authors (Required)):

The authors have done a good job at addressing the comments raised in the previous review and the paper is substantially improved. I had few major criticisms then and would be happy to see this accepted now.

1st Revision - Authors' Response to Reviewers: January 7, 2025

Reviewer #1 (Comments to the Authors (Required)):

The authors made satisfactory changes in response to the Reviewer Commons comments. They took these very seriously. The science is clearly presented, described, and claims made are well evidenced.

Reviewer #2 (Comments to the Authors (Required)):

I stand with my initial, very positive assessment during the Review Commons process, please see my original review report for specifics. In the meantime, the authors have fully and sufficiently addressed my comments during the revision and the resulting final version of the paper is a very strong manuscript that should be accepted without the need of further changes and in my opinion should be highlighted by a specialist in the field, since it comprehensively addresses the very important point of signalling lipid buffering by widely used biosensors in a quantitative fashion.
André Nadler

Reviewer #3 (Comments to the Authors (Required)):

The authors have done a good job at addressing the comments raised in the previous review and the paper is substantially improved. I had few major criticisms then and would be happy to see this accepted now.

Response:

We are grateful to all three reviewers for their comprehensive, constructive comments – they greatly improved the manuscript. We are thrilled that we are all in agreement that the paper is ready for publication.

We have only made minor edits to the title and abstract to bring the length into compliance with the journal's limits.